# Compositional Behavioral Semantics
# for State Abstraction in Reinforcement Learning

**Yivan Zhang** [1 2]  **Ziyan "Ray" Luo** [3 4]  **Manuel Baltieri** [5 6]

## Abstract

State abstraction plays a key role in scaling re-
inforcement learning to complex but structured
systems. In studying such systems, a wide range
of behavioral structures have been studied in re-
inforcement learning, including value functions,
invariants, bisimulation relations, and behavioral
metrics. However, a general principle for deter-
mining what structures are provably preserved un-
der state abstraction is still lacking. In this paper,
we present a unified framework for defining and
analyzing behavioral structures in reinforcement
learning. Our framework provides a composi-
tional way to specify behavioral semantics based
on local, one-step descriptions of system dynam-
ics. Using this framework, we establish results
showing how behavioral structures can be safely
transferred between abstract and concrete systems.
We further show how to construct quantitative
metrics from logical behavioral semantics with
soundness guarantees. Together, these results pro-
vide a principled foundation for reasoning about
behaviors under state abstraction in reinforcement
learning and offer reusable definition and proof
principles for a broad class of behavioral struc-
tures in reinforcement learning.

## 1. Introduction

*Reinforcement learning* (RL) (Sutton & Barto, 1998) has
been applied to increasingly complex domains such as
robotics and finance (Kober et al., 2013; Tang et al., 2025;
Moody et al., 1998; Liu et al., 2024), where large or high-
dimensional state spaces of environments make learning and
planning directly over raw states computationally intractable.
*State abstraction* (Abel, 2022) manages this complexity by
mapping concrete states to abstract states while preserving
decision-relevant behavioral properties (Mohan et al., 2024;
Echchahed & Castro, 2025).

State abstraction has been studied under a variety of forms,
differing in how states are grouped and what aspects of
behavior are preserved. In modern RL, state abstraction
is often realized by learning representations that compress
high-dimensional observations into latent states for control
and prediction (Echchahed & Castro, 2025). A prominent
family is based on *bisimulation equivalences and metrics*
(Dean & Givan, 1997; Givan et al., 2003; Ferns et al., 2004;
2011), originating in the analysis of concurrent systems and
system semantics (Milner, 1989) and later adapted to RL in
many variants (Taylor et al., 2008; Castro, 2020; Zhang et al.,
2021; Castro et al., 2021; Tao et al., 2025). A closely related
approach trains encoders to preserve rewards and one-step
dynamics, as in *model-irrelevance abstraction* (Li et al.,
2006), *self-predictive abstraction* (Schwarzer et al., 2021; Ni
et al., 2024; Voelcker et al., 2024), among others (François-
Lavet et al., 2019; Gelada et al., 2019). Weaker notions,
such as *value equivalence* (Givan et al., 2003; Li et al., 2006;
Grimm et al., 2020), only identify states that yield similar
values. However, in practice, state abstraction methods often
combine multiple objectives, making it difficult to isolate
individual abstraction requirements (Ni et al., 2024; Luo
et al., 2025).

Despite their differences, these state abstraction criteria can
be understood by asking which aspects of an interactive sys-
tem's behavior are meant to be preserved. This perspective
shifts the focus from defining abstractions themselves to un-
derstanding their behavioral consequences. The preserved
properties may be unary or binary, logical or quantitative,
and exact or approximate. To study them uniformly, we
need precise notions of *behavioral structure* and principled
ways to compare behaviors across states and systems. As a
salient example, the *value function* can be viewed as a unary,
quantitative behavioral measure that summarizes expected
returns over time (Sutton & Barto, 1998). On the logical
side, safe RL uses temporal logic specifications (Alshiekh
et al., 2018) and various other constraint formulations (Hsu

---

[1]The University of Tokyo [2]RIKEN AIP [3]Mila - Quebec
Artificial Intelligence Institute [4]McGill University [5]Araya Inc.,
Tokyo, Japan [6]University of Sussex. Correspondence to: Yi-
van Zhang <yivan.zhang@k.u-tokyo.ac.jp>, Ziyan "Ray" Luo
<ziyan.luo@mail.mcgill.ca>.

*Proceedings of the 43rd International Conference on Machine
Learning*, Seoul, South Korea. PMLR 306, 2026. Copyright 2026
by the author(s).

et al., 2021; Yu et al., 2022; Wachi et al., 2024) to define unary behavioral requirements, while the bisimulation relation (Dean & Givan, 1997) is a canonical example of a binary behavioral structure. On the quantitative side, safety value functions are unary behavioral quantities for measuring risk (Wachi et al., 2024), while various behavioral metrics are binary structures central to state abstraction (Luo et al., 2025). Despite their shared goal of characterizing behavior via dynamics, these notions remain fragmented and come with limited guarantees for how they transfer under state abstraction.

This fragmentation also manifests in a recurring pattern in RL theory, where even mild variants of existing behavioral structures are analyzed from scratch (Taylor et al., 2008; Castro, 2020; Zhang et al., 2021), frequently following the same proof template (Ferns et al., 2004). This pattern suggests the existence of shared definition and proof principles, but the RL literature lacks a unified formal toolkit that makes these arguments reusable across behavioral structures. Our goal is to make such shared principles explicit by providing a compositional framework for defining behavioral structures and analyzing their transfer through state abstraction.

In this work, we present a compositional framework for defining and analyzing behavioral structures in RL through the lens of *coalgebras* (Rutten, 2000; Jacobs, 2016), which provide a uniform language for modeling transition systems and their behaviors. The main contributions are threefold:

- **A compositional definition of behavioral structures.** We introduce a uniform definition of behavioral structures that subsumes invariants, value functions, and bisimulation relations or metrics as instances of a shared compositional construction, replacing many case-specific definitions with a common formal recipe.
- **General transfer guarantees for state abstraction.** We model state abstraction as *coalgebra homomorphisms* and prove that behavioral structures transport soundly along such abstractions via the pullback and pushforward operations between the concrete and abstract systems.
- **A general bridge from logical to quantitative semantics.** Under an explicit algebraic compatibility condition, we show how quantitative metrics can be constructed from logical behavioral semantics systematically.

Concretely, Section 2 introduces a coalgebraic framework for *behavioral systems*, specifying the systems considered in this work. Section 3 develops the core technical tools consisting of *bundles*, *liftings*, and the *pullback* and *pushforward* operations, which we use to formally define and analyze *behavioral structures*. Section 4 instantiates the abstract framework with some existing semantics in RL, illustrating its generality and expressiveness. A rigorous description of the theory is deferred to the appendix.

## 2. Behavioral Systems

In this section, we introduce a coalgebraic framework for modeling behavioral systems. We formalize state transition systems as *coalgebras* in Section 2.1, define state abstraction as a *homomorphism* between systems of the same type in Section 2.2, and formulate policy-dependent transition as a *natural transformation* between systems of different types in Section 2.3. We will use two system types throughout: stochastic Moore machines for environments, and hidden Markov models for the policy-conditioned systems obtained by composing an environment with a policy.

### 2.1. State transition system

An RL *environment* is usually modeled as a Markov decision process (MDP) (Puterman, 1994) or a partially observable MDP (POMDP) (Kaelbling et al., 1998) with potentially infinite spaces of *states* $S$, *actions* $A$, and *observations* $O$, and a *transition* map $t : S \times A \to \mathbb{P}S$,[1] an *observation* map $o : S \to O$, and an optional notion of initialization and termination. The observation space $O$ can be the space $S \times \mathbb{R}$ of the states themselves and scalar rewards for an MDP. An RL *agent* interacts with the environment by choosing actions based on observations to maximize an optimality criterion, e.g., the discounted sum of rewards (Sutton & Barto, 1998).

In this paper, we work with the same ingredients, but with an explicit *input-output interface* between the environment and agent, which allows us to analyze the one-step behaviors of each component separately and then combine them to understand the overall system *compositionally*. This focus on explicit interface specification and composition aligns with recent research on agents and RL (Abel et al., 2023; Mohan et al., 2024; Abel et al., 2025; Rosas et al., 2025; Boyd et al., 2025).

A core concept in the coalgebraic framework is that of a *functor* (Mac Lane, 1978), which is informally defined as follows (see Appendix A.1 for a rigorous definition):

**Definition 2.1** (Functor)**.** A *functor* $F$ is a map assigning to each set $X$ a set $FX$ and to each map $f : X \to Y$ a map $Ff : FX \to FY$, preserving identity and composition.

A functor is a "*type constructor*" that defines the system type. Given a functor $F$, we define state transition systems as $F$-*coalgebras* (Rutten, 2000; Jacobs, 2016):

**Definition 2.2** (System)**.** A *state transition system* is an $F$-*coalgebra*, i.e., a set $X$ with a map $t_X : X \to FX$.

---

[1]For a set $X$, $\mathbb{P}X$ denotes the set of probability distributions with finite support on $X$, and a stochastic map $p : X \to \mathbb{P}Y$ can be equivalently viewed as a function $X \times Y \to [0, 1]$ that maps each pair of input and output to the probability of the output given the input. We also use the usual probability notation $p(y|x) \in \mathbb{P}Y$ to denote the output of a map $p : X \to \mathbb{P}Y$ at the input $x \in X$.

An $F$-coalgebra $t_X$ describes a specific one-step behavior of type $F$ on the state space $X$. Following Rutten (2000), we use *coalgebra* and *system* synonymously.

Next, we introduce two concrete examples commonly used in RL to instantiate the above definitions.

First, we focus solely on the environment and model its dynamics as a *stochastic Moore machine* (Rutten, 2000; Silva et al., 2013).[2] Informally, a Moore machine is a state machine whose current state emits an output while the input determines the next-state transition. In our RL setting, the input is an action and the output is an observation. We can specify the system type via a functor and define a transition system as follows:

**Example 2.3** (Stochastic Moore machine functor). The functor $F_{\text{Moore}}$ for *stochastic Moore machines* is given by

$$F_{\text{Moore}}X := (\mathbb{P}X)^A \times O, \tag{1}$$

$$F_{\text{Moore}}f := (\mathbb{P}f)^A \times \text{id}_O, \tag{2}$$

where $\mathbb{P}X$ is the set of probability distributions with finite support on $X$, and $\mathbb{P}f : \mathbb{P}X \to \mathbb{P}Y$ is the pushforward of distributions on $X$ through $f$; $(\mathbb{P}X)^A$ is the set of maps from $A$ to $\mathbb{P}X$, and $(\mathbb{P}f)^A : (\mathbb{P}X)^A \to (\mathbb{P}Y)^A$ is the postcomposition of maps from $A$ to $\mathbb{P}X$ with $\mathbb{P}f$; $\text{id}_O$ is the identity map. In other words, $F_{\text{Moore}}X$ has elements $(p : A \to \mathbb{P}X, o \in O)$; $F_{\text{Moore}}f : F_{\text{Moore}}X \to F_{\text{Moore}}Y$ maps $(p, o)$ to $(f_*p, o)$, where $f_*p : A \xrightarrow{p} \mathbb{P}X \xrightarrow{\mathbb{P}f} \mathbb{P}Y$ maps each action $a \in A$ to the pushforward $f_*p(a) \in \mathbb{P}Y$ of the distribution $p(a) \in \mathbb{P}X$ through $f$.

**Example 2.4** (Stochastic Moore machine). A *stochastic Moore machine* is a system on a set $X$ of type $F_{\text{Moore}}$ in Example 2.3:

$$\langle t_X, o_X \rangle : X \to \quad (\mathbb{P}X)^A \quad \times \quad O \\ x \mapsto \big( [a \mapsto t_X(x'|x, a)] , o_X(x) \big) \tag{3}$$

where $t_X : X \to (\mathbb{P}X)^A$ is the *transition* map, mapping each state to a map from actions to next state distributions, and $o_X : X \to O$ is the *observation* map, mapping each state to an observation.[3]

The functor $F_{\text{Moore}}$ captures the *input-output interface* of an RL environment, where the input is an action from $A$ and

the output is an observation from $O$ (cf. Abel et al. (2025)). It also captures the *dependency structure* between the next state distribution and the current state and action, while the observation only depends on the current state. It does not include an optimality criterion (e.g., the discount factor for reward aggregation), which we believe should be an agent-side design, nor the episode initialization and termination, which appear in some formulations of MDPs or POMDPs (Puterman, 1994; Kaelbling et al., 1998).

Next, we consider a *hidden Markov model* (Rabiner, 1989), a system without inputs:

**Example 2.5** (Hidden Markov model functor). The functor $F_{\text{Markov}}$ for *hidden Markov models* is given by

$$F_{\text{Markov}}X := \mathbb{P}X \times O, \tag{4}$$

$$F_{\text{Markov}}f := \mathbb{P}f \times \text{id}_O . \tag{5}$$

Similar to Example 2.4, a hidden Markov model is a system of type $F_{\text{Markov}}$ in Example 2.5, except that the transition map $t_X : X \to \mathbb{P}X$ does not depend on actions. We omit the explicit definition here for brevity. In RL, composing an environment modeled as a stochastic Moore machine with an agent results in a closed-loop system parameterized by the policy, which can be modeled as a hidden Markov model. We discuss this composition in Section 2.3.

### 2.2. Homomorphism between systems of the same type

A central and deliberately strong notion of state abstraction is to preserve the dynamics of the original system in the abstract system. We show that this requirement can be formalized as a *coalgebra homomorphism* (Rutten, 2000; Jacobs, 2016) between two systems of the same type:

**Definition 2.6** (Coalgebra homomorphism). Given two $F$-coalgebras $t_X : X \to FX$ and $t_Y : Y \to FY$, a map $f : X \to Y$ is an $F$-*coalgebra homomorphism* if the following diagram commutes:

$$\begin{array}{ccc} X & \xrightarrow{\ f\ } & Y \\ t_X \downarrow & & \downarrow t_Y \\ FX & \xrightarrow{\ Ff\ } & FY \end{array} \tag{6a}$$

which means the following equation holds:

$$t_Y \circ f = Ff \circ t_X. \tag{6b}$$

A homomorphism is a dynamic-preserving map between two systems without changing their input-output interfaces. For stochastic Moore machines in Example 2.4, Def. 2.6 is instantiated as follows:

$$\begin{array}{ccc} X & \xrightarrow{\qquad\qquad f \qquad\qquad} & Y \\ \langle t_X, o_X \rangle \downarrow & & \downarrow \langle t_Y, o_Y \rangle \\ (\mathbb{P}X)^A \times O & \xrightarrow{(\mathbb{P}f)^A \times \text{id}_O} & (\mathbb{P}Y)^A \times O \end{array} \tag{7a}$$

$$\langle t_Y, o_Y \rangle \circ f = \big( (\mathbb{P}f)^A \times \text{id}_O \big) \circ \langle t_X, o_X \rangle, \tag{7b}$$

---

[2]We choose "*stochastic transition and deterministic action-independent observation*" for simplicity. More complex systems such as "*stochastic and correlated transition and observation with termination*" can be modeled coalgebraically in a similar way (Silva et al., 2013; Jacobs, 2016). See Appendix A.2 for details.

[3]For two functions $f : C \to A$ and $g : C \to B$, their *pairing* $\langle f, g \rangle : C \to A \times B$ is the unique function that applies these two functions to the same input, mapping an input $c \in C$ to a pair $(f(c), g(c)) \in A \times B$ of outputs.

Note that the transition map $t_X : X \to (\mathbb{P}X)^A$ is isomorphic to a binary map $t_X : X \times A \to \mathbb{P}X$ via *currying*, which, with slight abuse of notation, we also denote by the same symbol.

which is further equivalent to the following two equations:[4]

$$\forall x \in X. \ \forall a \in A. \ \mathsf{t}_Y(f(x), a) =_{\mathbb{P}Y} f_* \mathsf{t}_X(x, a), \quad (7\text{c.i})$$

$$\forall x \in X. \ \mathsf{o}_Y(f(x)) =_O \mathsf{o}_X(x). \quad (7\text{c.ii})$$

We can spell out Eq. (7c.i) using pushforward as follows:

$$\forall y' \in Y. \ \mathsf{t}_Y(y'|f(x), a) = \sum_{x' \in f^{-1}(y')} \mathsf{t}_X(x'|x, a), \quad (7\text{c.i}')$$

where $f^{-1}(y') := \{x' \in X \mid f(x') = y'\} \subseteq X$ is the fiber (inverse image) of $y' \in Y$ through $f : X \to Y$.

The homomorphism requirements in Eq. (7) have been used for state abstraction in prior works under various other names (Li et al., 2006; François-Lavet et al., 2019; Schwarzer et al., 2021; Subramanian et al., 2022). Recently, Ni et al. (2024); Voelcker et al. (2024); Luo et al. (2025) formalized and analyzed these requirements theoretically and empirically. Specifically, Eq. (7c.i) is the *next latent state distribution prediction* condition (Ni et al., 2024, ZP), while Eq. (7c.ii) is the *reward prediction* condition (Ni et al., 2024, RP) when the observation maps are reward functions. Under the coalgebraic framework, Eq. (7) thus unifies these two properties as a single homomorphism condition. Such a form of state abstraction can be defined as follows:

**Definition 2.7** (Homomorphic abstraction). Given a system $S \to FS$ on a state space $S$, an encoder $\phi : S \to Z$ to a representation space $Z$ *abstracts the system* if there exists an abstract system $Z \to FZ$ on the representation space $Z$ such that $\phi$ is a homomorphism as in Def. 2.6.

Note that this dynamic-preservation is a strong requirement and is not satisfied by all abstractions. Frequently, we may only want to preserve certain structures or properties of the original system, such as value functions, behavioral metrics, or optimal policies (Li et al., 2006; Grimm et al., 2020; Luo et al., 2025). We discuss such behavioral structures and how they transfer through homomorphisms in Section 3.

A relevant yet different notion is the *MDP homomorphism* (Ravindran & Barto, 2001; 2002) in *state-action abstraction*, which involves an interface change. We discuss how our framework can accommodate this notion in Appendix F.

## 2.3. Transformation between systems of different types

In RL, we often need to relate systems of different types, for example, when an agent interacts with an environment to form a closed-loop system, or when the environment restricts the available actions or transforms its outputs. We therefore turn to *natural transformations* as a principled way to compare and connect such heterogeneous systems (Rutten, 2000, Chapter 15):

---

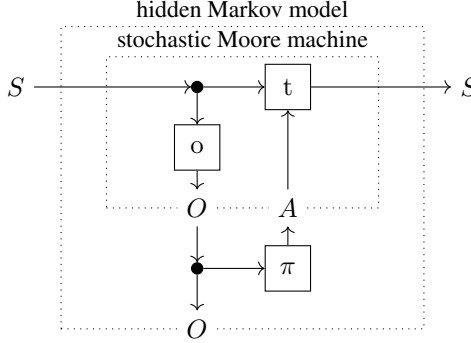

*Figure 1.* Policy-dependent transition obtained by closing the loop between the observation and action spaces via a policy $\pi : O \to A$. $\mathsf{t} : S \times A \to \mathbb{P}S$ is the stochastic transition map of the environment, $\mathsf{o} : S \to O$ is the observation map, and dots $\bullet$ represent copying.

**Definition 2.8** (Natural transformation). Let $F$ and $G$ be two functors. A *natural transformation* $\alpha : F \Rightarrow G$ is a family of maps $\alpha_X : FX \to GX$ such that for every map $f : X \to Y$, the following diagram commutes:

$$
\begin{array}{ccc}
FX & \xrightarrow{Ff} & FY \\
\alpha_X \downarrow & & \downarrow \alpha_Y \\
GX & \xrightarrow{Gf} & GY
\end{array}
\quad (8\text{a})
$$

which means the following equation holds:

$$\alpha_Y \circ Ff = Gf \circ \alpha_X. \quad (8\text{b})$$

An important fact is that we can use a natural transformation $\alpha : F \Rightarrow G$ to transform an $F$-coalgebra $t_X : X \to FX$ into a $G$-coalgebra $\alpha_X \circ t_X : X \xrightarrow{t_X} FX \xrightarrow{\alpha_X} GX$ on the same state space $X$, and this transformation preserves homomorphisms. This transformation can be seen as the composition of the diagrams in Eqs. (6a) and (8a).

A key example is the transformation from an environment dynamic (modeled as a stochastic Moore machine) to an environment-agent interaction (modeled as a hidden Markov model):

**Example 2.9** (Policy-dependent transition as a natural transformation). Let $\pi : O \to A$ be a policy. We can define a natural transformation $\alpha^\pi : F_{\text{Moore}} \Rightarrow F_{\text{Markov}}$ from $F_{\text{Moore}}$ in Example 2.3 to $F_{\text{Markov}}$ in Example 2.5:

$$
\alpha_X^\pi : \quad \begin{array}{ccc} F_{\text{Moore}}X & \to & F_{\text{Markov}}X \\ \begin{pmatrix} p : A \to \mathbb{P}X \\ o \in O \end{pmatrix} & \mapsto & \begin{pmatrix} p(x|\pi(o)) \in \mathbb{P}X \\ o \in O \end{pmatrix} \end{array} \quad (9)
$$

We can verify that $\alpha^\pi$ satisfies the naturality condition in Def. 2.8. An illustration is shown in Fig. 1.

This construction models the closed-loop system induced by an environment and an agent following a stationary policy, which is useful for describing policy-dependent bisimulation (Castro, 2020) and value equivalence (Li et al., 2006; Grimm et al., 2020).

*Table 1.* Examples of $n$-ary bundles $X^n \to V$

|  | $n = 1$ | $n = 2$ |
|---|---|---|
| truth values $V = \{\top, \bot\}$ | predicate $p : X \to \{\top, \bot\}$ | relation $r : X \times X \to \{\top, \bot\}$ |
| real values $V = [0, \infty]$ | quantity $q : X \to [0, \infty]$ | distance $d : X \times X \to [0, \infty]$ |

*Table 2.* Examples of combinators, aggregators, and barycenters

| **combinator** | $\oplus : V \times V \to V$ |
|---|---|
| conjunction | $\wedge : \{\top, \bot\} \times \{\top, \bot\} \to \{\top, \bot\}$ |
| addition | $+ : [0, \infty] \times [0, \infty] \to [0, \infty]$ |
| **aggregator** | $\Box_X : V^X \to V$ |
| universal quantifier | $\forall_X : \{\top, \bot\}^X \to \{\top, \bot\}$ |
| summation | $\sum_X : [0, \infty]^X \to [0, \infty]$ |
| **barycenter** | $\odot : \mathbb{P}V \to V$ |
| almost surely | $\forall : \mathbb{P}\{\top, \bot\} \to \{\top, \bot\}$ |
| expectation | $\mathbb{E} : \mathbb{P}[0, \infty] \to [0, \infty]$ |

# 3. Behavioral Semantics

Having formalized how state transition systems generate behavior, we next develop a framework for specifying and reasoning about structures on the state space of such systems that determine how behavior is interpreted. In Sections 3.1 to 3.4, we present the notions of *bundle*, *lifting*, *pullback*, and *pushforward*, which form the technical foundation for formally defining *behavioral structures*. In Section 3.5, we present theoretical results regarding behavioral structure transfer to justify our framework. We use safety predicates, value functions, and bisimulation relations and metrics as running examples in this section.

## 3.1. Bundle

First, we introduce the concept of a *bundle*, which subsumes structures such as predicates, quantities, relations, distances, and, more generally, $n$-ary structures over the state space:

**Definition 3.1** (Bundle). Given an arity $n \in \mathbb{N}$, an *$n$-ary bundle* over a set $V$ is a set $X$ with a map $h_X : X^n \to V$.

In this paper, we mainly focus on unary bundles ($n = 1$), binary bundles ($n = 2$), logical definitions ($V = \{\top, \bot\}$), and quantitative measurements ($V = [0, \infty]$ or $\mathbb{R}$), with examples summarized in Table 1. We equip $V$ with a pre-order $\preceq$ to compare bundles, e.g., $\bot \to \top$ for $\{\top, \bot\}$ and $\geq$ for $[0, \infty]$ or $\mathbb{R}$. Note that not all structures are compatible with behavioral systems. We will discuss which structures qualify as *behavioral structures* in Section 3.3.

## 3.2. Lifting of bundles

Given a state space $X$ and a one-step behavior space $FX$ specified by a functor $F$, we want to systematically construct a structure on $FX$ from a structure on $X$. This is where *lifting* comes in: *defining by induction* on the structure of $F$ (Kurz, 2001; Pattinson, 2003; Jacobs, 2016). Specifically, a *bundle lifting* $\lambda_X$ is a map from a bundle $h_X : X^n \to V$ to a bundle $h_{FX} : (FX)^n \to V$. We defer the rigorous definition to Appendix B and only present some concrete procedures for the functors in Examples 2.3 and 2.5 here. We give examples for $n = 1$ and $n = 2$ separately, while keeping the codomain $V$ generic. We will use the operators in Table 2, which correspond to the three components of the system (product, exponentiation, and probability) in Examples 2.3 and 2.5.

**Unary bundle lifting** We begin with the relatively simple case of $n = 1$:

**Example 3.2** (Unary bundle lifting for $F_{\text{Markov}}$). Given the operators in Table 2 and a unary bundle $h_O : O \to V$ on observations, a *unary bundle lifting* $\lambda_X$ for the functor $F_{\text{Markov}}$ in Example 2.5 is given by

$$h_{FX}(p, o) := (\odot\, h_{X*}p) \oplus h_O(o). \qquad (10)$$

Here, $h_{X*}p \in \mathbb{P}V$ is the usual pushforward of a state distribution $p \in \mathbb{P}X$ through the map $h_X : X \to V$.

The following three concrete examples of logical, real-valued, and vector-valued bundles instantiate this pattern in the RL context:

**Example 3.3** (Safety property). Let $V = \{\top, \bot\}$, and let $h_O : O \to \{\top, \bot\}$ be a safety predicate on observations. The following lifting maps a predicate $h_X : X \to \{\top, \bot\}$ to an observation-gated safety property:

$$h_{FX}(p, o) := \big(\forall_{x \sim p} h_X(x)\big) \wedge h_O(o), \qquad (11)$$

which means that the observation $o$ must be safe ($h_O(o)$), and ($\oplus = \wedge$) the predicate $h_X$ holds almost surely ($\odot = \forall$) over the state distribution $p$.

This corresponds to a one-step local form of almost-sure safety or reach-avoid constraints in safe RL (Alshiekh et al., 2018; Hsu et al., 2021; Yu et al., 2022; Wachi et al., 2024).

**Example 3.4** (Reward aggregation). Let $V = \mathbb{R}$, and let $h_O : O \to \mathbb{R}$ be a reward function on observations. The following lifting maps a quantity $h_X : X \to \mathbb{R}$ to a reward aggregation scheme:

$$h_{FX}(p, o) := \gamma \underset{x \sim p}{\mathbb{E}}\, h_X(x) + h_O(o), \qquad (12)$$

where $\gamma \in [0, 1]$ is a discount factor.

The reward aggregation scheme in Example 3.4 does not rely on $h_O : O \to \mathbb{R}$ being a task reward. If $h_O$ is instead an arbitrary scalar *cumulant*, such as a sensor reading, an

event indicator, or a component of a feature vector, the same lifting specifies the aggregation scheme underlying a *general value function* (Sutton et al., 2011).

**Example 3.5** (Successor feature). Let $h_O : O \xrightarrow{\phi} \Phi \xrightarrow{w} \mathbb{R}$ be a scalar cumulant that factors through a vector space of features $\Phi = \mathbb{R}^d$, where $\phi$ is a feature map and $w$ is a linear map. Taking $V = \Phi$ gives the vector-valued aggregation scheme for for a vector-valued bundle $h_X^\Phi : X \to \Phi$:

$$h_{FX}^\Phi(p, o) := \gamma \mathop{\mathbb{E}}_{x \sim p} h_X^\Phi(x) + \phi(o). \qquad (13)$$

For a real-valued bundle $h_X = w \circ h_X^\Phi$, linearity gives $w(h_{FX}^\Phi(p, o)) = h_{FX}(p, o)$, where $h_{FX}$ is the real-valued aggregation in Example 3.4.

This is the one-step vector aggregation scheme behind the idea of *successor features* (Dayan, 1993; Barreto et al., 2017; Machado et al., 2023; Wiltzer et al., 2024).

These examples demonstrate that *choosing the operators amounts to choosing the semantics* of the lifted bundle. The bundle arity, codomain, and these design choices together determine the behavioral structure under study. For example, the almost-sure operator in Example 3.3 defines a more stringent semantics.[5] The addition in Example 3.4 can be replaced with other combinators such as maximum, yielding different reward semantics.

**Binary bundle lifting**  When $n = 2$, the design space of bundle liftings is richer. A common choice is the following:

**Example 3.6** (Binary bundle lifting for $F_{\text{Moore}}$). Given the operators in Table 2 and a binary bundle $h_O : O \times O \to V$ on observations, a *binary bundle lifting* $\lambda_X$ for the functor $F_{\text{Moore}}$ in Example 2.3 is defined inductively as follows:

$$h_{FX}((p, o), (p', o')) := h_{(\mathbb{P}X)^A}(p, p') \oplus h_O(o, o'), \quad (14)$$

$$h_{(\mathbb{P}X)^A}(p, p') := \mathop{\square}_{a \in A} h_{\mathbb{P}X}(p(a), p'(a)), \qquad (15)$$

$$h_{\mathbb{P}X}(\mu, \mu') := \mathop{\vee}_{\pi \in \Pi_X(\mu, \mu')} \odot h_{X_*}\pi, \qquad (16)$$

where $\Pi_X(\mu, \mu') \subset \mathbb{P}(X \times X)$ is the set of all *couplings* of $\mu, \mu' \in \mathbb{P}X$, and $\vee$ is the *join* operator for $(V, \preceq)$, i.e., $\exists$ for $\{\top, \bot\}$ and inf for $[0, \infty]$.

It is noteworthy that the lifting operation does not depend on a specific system (i.e., a coalgebra), but solely on the system type (i.e., the functor). Next, we introduce the *pullback* and *pushforward* operations, which allow us to *transfer bundles* between different spaces along maps.

## 3.3. Pullback of bundles

The pullback operation transfers bundles *contravariantly* from a target space $Y$ to a source space $X$ along a map $f : X \to Y$:

**Definition 3.7** (Pullback). Given a map $f : X \to Y$, the *pullback* of a bundle $h_Y : Y^n \to V$ along $f$ is the bundle $f^*h_Y : X^n \to V := h_Y \circ f^n$ given by *precomposition*.

**Pullback along a system**  In Section 3.2, we have defined a bundle lifting $\lambda_X$ that constructs a bundle $h_{FX}$ on $FX$ from a bundle $h_X$ on $X$. Given a system $t_X : X \to FX$, it is natural to pull the lifted bundle $h_{FX}$ back along $t_X$ to obtain a new bundle on $X$, so that we can compare it with the original bundle $h_X$ on the same space:

**Definition 3.8** (Closure operator). Given an $F$-coalgebra $t_X : X \to FX$ and a bundle lifting $\lambda_X$, a *closure operator* is defined as the composition $T_X := t_X^* \circ \lambda_X$. In other words, for any bundle $h_X : X^n \to V$ and $x_1, \ldots, x_n \in X$, $T_X(h_X)(x_{1:n}) := \lambda_X(h_X)(t_X(x_1), \ldots, t_X(x_n))$.[6]

We call a bundle $h_X$ a *behavioral structure*, as in the sense of system behavior specification (Reichel, 1981; Jacobs, 2016), if it is a post-fixed point or a fixed point of a closure operator $T_X$:

**Definition 3.9** (Behavioral structure). A *behavioral structure* $h_X$ is a *post-fixed point* of the closure operator $T_X$, i.e., $h_X \preceq T_X(h_X)$ or a *fixed point*, i.e., $h_X = T_X(h_X)$.

This pattern is ubiquitous in RL. For example, fixing a policy $\pi$, the closure operator for the reward aggregation lifting in Example 3.4 is exactly the *Bellman operator* $\mathcal{B}$ on quantities $v : X \to \mathbb{R}$ (Sutton & Barto, 1998):

$$\mathcal{B}(v)(x) := \gamma \mathop{\mathbb{E}}_{x' \sim t_X^\pi(x)} v(x') + h_O(\mathsf{o}_X(x)), \qquad (17)$$

where $\langle \mathsf{t}_X^\pi, \mathsf{o}_X \rangle : X \to \mathbb{P}X \times O$ is the policy-dependent transition as in Example 2.9, and a fixed point of $\mathcal{B}$ is the state value function $v^\pi$. In state abstraction, other examples include *bisimulation relations* (Dean & Givan, 1997; Givan et al., 2003) and *bisimulation metrics* (Ferns et al., 2004; 2011), which we will spell out in Section 4.

**Pullback along an encoder**  Another use of pullback is to relate bundles on the representation space $Z$ with those on the concrete state space $S$ along an encoder $\phi : S \to Z$. For example, Luo et al. (2025, Definition 1) conceptually unified and analyzed several state representation learning objectives by requiring the encoder to be an isometry, i.e., $d_S = \phi^* d_Z$, where $d_S$ is a behavioral metric on $S$ and $d_Z$ is a chosen metric on $Z$, typically reflecting its geometry. This allows us to verify whether $d_Z$ encodes the desired behaviors. As pullback is simply precomposition, we can

also pull back other structures. For example, given a policy $\pi_Z : Z \to A$ on the representation space $Z$, we can pull it back along an encoder $\phi : S \to Z$ to obtain a policy $\phi^* \pi_Z : S \to A$ on the concrete state space $S$.

### 3.4. Pushforward of bundles

On the other hand, the pushforward operation constructs bundles *covariantly* from a source space $X$ to a target space $Y$ along a map $f : X \to Y$:

> **Definition 3.10** (Pushforward). Given a map $f : X \to Y$, the *pushforward* of a bundle $h_X : X^n \to V$ along $f$ is the bundle $f_* h_X : Y^n \to V$ given by
> $$f_* h_X(y_{1:n}) := \bigvee_{x_i \in f^{-1}(y_i)} h_X(x_{1:n}), \qquad (18)$$
> where $f^{-1}(y_i) := \{x_i \in X \mid f(x_i) = y_i\} \subseteq X$ is the fiber (inverse image) of $y_i \in Y$ through $f : X \to Y$, and $\vee$ is the join operator for $(V, \preceq)$. We assume that this join exists for the relevant fibers, including the empty-fiber case via the corresponding empty join.

In this work, the main use case is the pushforward along an encoder $\phi : S \to Z$: given a bundle $h_S : S^n \to V$ on the concrete state space $S$, the pushforward $\phi_* h_S : Z^n \to V$ produces a well-defined bundle on the representation space $Z$ by aggregating $h_S$ over all concrete realizations of a representation in a conservative manner.

### 3.5. Behavioral structure transfer

We now have all the necessary components: behavioral systems, maps between systems, behavioral structures, and operations for transferring structures. Finally, we present our theoretical results regarding behavioral structure transfer to show how these components fit together.

**Transfer of behavioral structures along homomorphisms**
First, we present the main theorems that justify the use of pullback and pushforward for transferring behavioral structures along homomorphisms. These theorems give general conditions under which familiar behavioral structures, including safety predicates, value functions, and bisimulation relations and metrics, can be pulled back or pushed forward across abstraction maps. Our results are based on the following lemma:

**Lemma 3.11.** *Given two systems* $(X, t_X)$ *and* $(Y, t_Y)$, *two liftings* $\lambda_X$ *and* $\lambda_Y$ *constructed with the same operators, and the induced closure operators* $T_X$ *and* $T_Y$, *if a map* $f : X \to Y$ *is a homomorphism, then*
$$T_X \circ f^* \preceq f^* \circ T_Y, \qquad (19)$$
$$f_* \circ T_X \preceq T_Y \circ f_*. \qquad (20)$$

This lemma shows that the liftings and closure operators are compatible with homomorphisms *by construction*. Based on

this lemma, we can prove the following behavioral structure transfer and construction theorems:

**Theorem 3.12** (Safe verification). *If* $\phi : (S, t_S) \to (Z, t_Z)$ *is a surjective homomorphism, then the pullback* $\phi^*$ *reflects post-fixed points: for any bundle* $h_Z : Z^n \to V$,
$$(h_Z \preceq T_Z(h_Z)) \leftarrow (\phi^* h_Z \preceq T_S(\phi^* h_Z)). \qquad (21)$$

This theorem states that if an encoder $\phi$ is *surjective* and preserves the dynamics, then any behavioral structure $h_Z$ on the abstract system can be safely verified on the concrete system by pulling it back along the encoder. The surjectivity condition, i.e., every abstract state corresponds to at least one concrete state, ensures that the abstract system has no extra states that could generate unverifiable behaviors.

**Theorem 3.13** (Safe construction). *If* $\phi : (S, t_S) \to (Z, t_Z)$ *is a homomorphism, then the pushforward* $\phi_*$ *preserves post-fixed points: for any bundle* $h_S : S^n \to V$,
$$(h_S \preceq T_S(h_S)) \to (\phi_* h_S \preceq T_Z(\phi_* h_S)). \qquad (22)$$

This theorem ensures that pushing forward any concrete behavioral structure along any homomorphism produces an abstract behavioral structure.

**Relation of behavioral structures over different values**
Next, we present a theorem that relates logical bundles ($V = \{\top, \bot\}$) and quantitative bundles ($V = [0, \infty]$), which allows us to systematically construct quantitative behavioral structures from logical ones:

**Theorem 3.14** (Relation of logical and quantitative bundles). *The zero predicate* $z : [0, \infty] \to \{\top, \bot\} := [x \mapsto (x = 0)]$ *maps a quantitative bundle* $h_X : X^n \to [0, \infty]$ *to a logical bundle* $z \circ h_X : X^n \to \{\top, \bot\}$. *If a quantitative lifting* $\lambda_X^{[0,\infty]}$ *and a logical lifting* $\lambda_X^{\{\top,\bot\}}$ *are constructed with operators in Table 2 such that the zero predicate* $z$ *is an algebra homomorphism between the operators,[7] then*
$$z \circ \lambda_X^{[0,\infty]}(h_X) = \lambda_X^{\{\top,\bot\}}(z \circ h_X). \qquad (23)$$

This theorem states that if the operators used in the liftings are homomorphic via the zero predicate, then the value of the quantitative bundle is zero if and only if the logical bundle evaluates to true. The homomorphism condition encodes the relationships between operators such as "*the sum/max of two non-negative numbers is zero if and only if both numbers are zero*" (combinator), or "*the expectation of a non-negative random variable is zero if and only if the variable is almost surely zero*" (barycenter). In this way, Theorem 3.14 is not only a relation between existing notions, but a construction principle: once a logical behavioral structure is specified as a bundle and the operators satisfy the homomorphism condition, the corresponding quantita-

---

[7]This means that (i) $z \circ \oplus_{[0,\infty]} = \oplus_{\{\top,\bot\}} \circ (z \times z)$, (ii) $z \circ \square_{[0,\infty]} = \square_{\{\top,\bot\}} \circ z^A$, and (iii) $z \circ \odot_{[0,\infty]} = \odot_{\{\top,\bot\}} \circ \mathbb{P}z$.

tive structure follows automatically. This generalizes the connection between bisimulation relations and bisimulation metrics (Ferns et al., 2004; 2011; Castro, 2020) to general behavioral structures.

# 4. State Abstraction in Reinforcement Learning

This section instantiates the framework on four familiar RL themes: next-observation prediction, model-irrelevance abstraction, bisimulation equivalences, and policy-dependent transition and value functions.

## 4.1. Next-observation prediction

First, we show that the *next-observation prediction* task used in Ota et al. (2020); Subramanian et al. (2022) follows from the homomorphism condition of the $F_{\mathrm{Moore}}$ type in Eq. (7):

**Proposition 4.1.** *If $f : X \to Y$ is a homomorphism from $\langle t_X, o_X \rangle$ to $\langle t_Y, o_Y \rangle$, then it is also a homomorphism from $(\mathbb{P}o_X)^A \circ t_X : X \to (\mathbb{P}O)^A$ to $(\mathbb{P}o_Y)^A \circ t_Y : Y \to (\mathbb{P}O)^A$, i.e., $\forall x \in X. \, \forall a \in A. \, o_{Y*}t_Y(f(x), a) =_{\mathbb{P}O} o_{X*}t_X(x, a)$.*

Since an $F_{\mathrm{Moore}}$-homomorphism preserves both the transition component in Eq. (7c.i) and the observation component in Eq. (7c.ii), it preserves next-observation distributions (see also Ni et al. (2024)). The converse need not hold, so next-observation prediction alone does not guarantee transition and observation preservation.

## 4.2. Model-irrelevance abstraction

Next, we incorporate the *model-irrelevance abstraction* (Li et al., 2006; Li, 2009) into our coalgebra framework and show that it is equivalent to the coalgebra homomorphism condition:

**Proposition 4.2.** *A model-irrelevance abstraction in Li et al. (2006) is exactly a coalgebra homomorphism in Def. 2.6.*

Concretely, a model-irrelevance abstraction requires that for an encoder $\phi : S \to Z$, if two states $s_1, s_2 \in S$ are mapped to the same representation, i.e., $\phi(s_1) =_Z \phi(s_2)$, then they must have the same observation and transition distributions, i.e., $o_S(s_1) =_O o_S(s_2)$ and for all actions $a \in A, \phi_*t_S(s_1, a) =_{\mathbb{P}Z} \phi_*t_S(s_2, a)$.

This is not written as a homomorphism, since it does not require a system on the representation space $Z$. It is instead the *kernel* view of a homomorphism, requiring $\phi$ to satisfy $\ker \phi \to \ker(\phi_*t_S) \wedge \ker o_S$, where the consequent is exactly $\ker(F_{\mathrm{Moore}}\phi \circ \langle t_S, o_S \rangle)$.[8] This implication holds exactly when $F_{\mathrm{Moore}}\phi \circ \langle t_S, o_S \rangle$ factors through $\phi$, which is the

homomorphism condition in Eq. (7). Thus Theorems 3.12 and 3.13 apply to model-irrelevance abstraction.

## 4.3. From homomorphism to bisimulation

In state abstraction, *bisimulation relations* are of particular interest (Dean & Givan, 1997; Givan et al., 2003). In our framework, we can describe a bisimulation relation for a system $t_X$ of type $F_{\mathrm{Moore}}$ as a post-fixed point of a closure operator in Def. 3.8 derived from a lifting in Example 3.6 where the combinator is conjunction $\wedge$, the aggregator is universal quantification $\forall$, and the barycenter operator is the almost-sure operator. This particular lifting is forced by the homomorphism requirement:

**Proposition 4.3.** *For an equivalence $r : X \times X \to \{\top, \bot\}$, let $q_r : X \to X/r$ be its quotient map. For an $F$-coalgebra $(X, t_X)$, define a monotone operator $T_X$ on equivalences as follows:*

$$T_X(r) := \ker(Fq_r \circ t_X) = t_X^* \ker(Fq_r). \quad (24)$$

*Then, the quotient map $q_r$ is a homomorphism if and only if the equivalence relation $r$ is a post-fixed point of $T_X$.*

In other words, the map from $r : X \times X \to \{\top, \bot\}$ to $\ker(Fq_r) : FX \times FX \to \{\top, \bot\}$ is exactly the lifting that characterizes whether the quotient map of an equivalence is a homomorphism. When we instantiate this result to the $F_{\mathrm{Moore}}$ functor, we get the usual definition of bisimulation equivalences. We defer the details to Appendix E.

Consequently, Propositions 4.2 and 4.3 together imply that model-irrelevance abstractions are quotient maps of bisimulation equivalences. This explains why self-prediction can recover bisimulation-based abstractions without explicitly computing bisimulation metrics, whose standard formulations require optimization over couplings (Ferns et al., 2004; Castro et al., 2021; Calo et al., 2024).

## 4.4. Value function and value equivalence

In Section 2.3, policy-dependent transition is modeled as a natural transformation from $F_{\mathrm{Moore}}$ in Example 2.3 to $F_{\mathrm{Markov}}$ in Example 2.5. Together with Proposition 4.3, this explains policy-dependent bisimulation equivalence (Castro, 2020) as the bisimulation structure obtained after closing the loop with a policy.

In Section 3.3, we showed that the Bellman operator is the closure operator induced by the reward aggregation lifting in Example 3.4 and a policy-dependent transition system of type $F_{\mathrm{Markov}}$, and the *value function* $v^\pi$ is a fixed point of this operator. Its kernel, defined by

$$(s \sim_v s') := (v^\pi(s) = v^\pi(s')), \quad (25)$$

is the *value equivalence* relation (Li et al., 2006; Li, 2009; Castro et al., 2009; Abel et al., 2016; 2020; Grimm et al.,

---

[8] The kernel $\ker f : X \times X \to \{\top, \bot\}$ of a map $f : X \to Y$ is the pullback $f^*(=_Y)$ of the equality $=_Y$ on the codomain.

2020). Unlike bisimulation, value equivalence need not preserve the one-step decomposition of behavior: equal values may result from different immediate cumulants and different future distributions that cancel after aggregation. Thus value equivalence is better viewed here as a relation derived from a unary behavioral structure, rather than as a behavioral relation specified directly by a one-step binary lifting.

## 5. Conclusion

**Summary**    We introduce a compositional foundation for defining and analyzing behavioral structures in RL, bringing unary invariants, value functions, and bisimulation relations or metrics into a single transferable framework. The system dynamics are modeled as coalgebras, a central and widely used notion of state abstraction is characterized as coalgebra homomorphisms, and behavioral structures are generated uniformly from system dynamics via a general lifting-and-pullback construction. We prove sound transfer via pullback and pushforward, and derive quantitative metrics from logical behavioral semantics.

**Compositionality**    In this paper, "*compositional*" means that global behavioral semantics are built by composing smaller components. At the system level, components such as transition, observation, and policy are composed into coalgebraic behavioral systems via *functor* constructions. At the semantics level, behavioral structures are defined via *lifting* using a wide choice of operators. At the abstraction level, these structures are composed with abstraction maps by pullback and pushforward, giving transfer results rather than case-specific proofs.

**Scope, limitations, and future work**    This work focuses on strict homomorphisms and exact structural preservation. A natural next step is to study relaxed commutation with explicit slack via *lax homomorphisms*. From there, one can develop a quantitative theory of approximation that turns this slack into weaker transfer guarantees, practical diagnostics, or training objectives. The bundle viewpoint also supports symbolic specifications, including safety and temporal logic constraints, although a fuller treatment lies beyond the scope of this paper. Another direction is to study weaker notions of *behavioral structure-preserving maps*, which may enable more flexible abstractions while retaining some behavioral guarantees. Finally, we have focused on the theoretical foundations of behavioral structures, but an important next step is to develop practical algorithms for learning and leveraging these structures in RL.

## Acknowledgements

We are grateful to Tianwei Ni for helpful discussions on representation learning in RL, and Doina Precup for valuable discussion and feedback. We thank the anonymous reviewers for their constructive feedback and suggestions.

YZ was supported by JSPS KAKENHI Grant Number JP26K21299. ZL was supported by the Canada CIFAR AI Chair Program and the R3AI program of the Canada First Research Excellence Fund. MB was supported by the Advanced Research + Invention Agency (ARIA) through project code MSAI-SE01-P011.

## Impact Statement

This paper develops theoretical tools intended to advance reinforcement learning. The results are primarily conceptual and methodological, and we do not anticipate immediate direct societal impacts.

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

# Contents

## Symbols

| Symbol | Name | Meaning |
|---|---|---|
| *Spaces* | | |
| $X, Y$ | generic spaces | Arbitrary sets used in the general definitions, which can be instantiated to $S$, $Z$, etc. |
| $S$ | state space | Concrete state space of the RL environment. |
| $Z$ | representation space | Abstract state or learned representation space. |
| $A$ | action space | Set of actions available to the agent. |
| $O$ | observation space | Set of observations emitted by the environment. |
| $V$ | value space | Codomain of bundles, typically $\{\top, \bot\}$, $[0, \infty]$, or $\mathbb{R}$. |
| $\mathbb{P}X$ | distributions on $X$ | Set of finitely supported probability distributions on $X$. |
| *Systems and dynamics* | | |
| $F$ | functor, system type | Endofunctor specifying the one-step behavior space $FX$. |
| $F_{\mathrm{Moore}} : X \mapsto (\mathbb{P}X)^A \times O$ | stochastic Moore type | Environment type $(\mathbb{P}X)^A \times O$. |
| $F_{\mathrm{Markov}} : X \mapsto \mathbb{P}X \times O$ | hidden Markov type | Policy-conditioned type $\mathbb{P}X \times O$. |
| $t_X : X \to FX$ | system | $F$-coalgebra describing one-step behavior on $X$. |
| $\mathrm{t}_X : X \times A \to \mathbb{P}X$ | transition map | Maps a state and action to a next-state distribution. |
| $\mathrm{o}_X : X \to O$ | observation map | Maps a state to its emitted observation. |
| *Abstraction and agents* | | |
| $\pi : O \to A$ | policy | Agent map from observations to actions. |
| $\phi : S \to Z$ | encoder | State abstraction map. |
| *Behavioral structures* | | |
| $h_X : X^n \to V$ | $n$-ary bundle | $n$-ary structure on $X$, such as a predicate, value, relation, or metric. |
| $\preceq\, : V \times V \to \{\top, \bot\}$ | preorder | Order used to compare bundles pointwise. |
| $\lambda_X : h_X \mapsto h_{FX}$ | lifting | Constructs a bundle on $FX$ from a bundle on $X$. |
| $f^* : h_X \mapsto h_Y$ | pullback | Contravariant transport of bundles by precomposition along $f : X \to Y$. |
| $f_* : h_X \mapsto h_Y$ | pushforward | Covariant transport of bundles by joining over fibers of $f : X \to Y$. |
| $T_X : h_X \mapsto t_X{}^* \circ h_{FX}$ | closure operator | Composition of a pullback and a lifting whose post-fixed points are behavioral structures. |
| *Operators for defining liftings* | | |
| $\oplus : V \times V \to V$ | combinator | Operator combining multiple values, e.g., conjunction or addition. |
| $\Box_X : V^X \to V$ | aggregator | Operator aggregating over input-indexed components, e.g., $\forall$ or $\sum$. |
| $\odot : \mathbb{P}V \to V$ | barycenter | Operator aggregating over probability distributions, e.g., almost-sure truth or expectation. |

## List of Figures

## List of Tables

# A. Preliminaries

In this section, we recall some basic definitions and key results in *category theory* that we will use throughout the development of our theory. Readers are referred to standard textbooks for more details (Mac Lane, 1978; Awodey, 2010).

## A.1. Category, functor, and natural transformation

The central categorical concepts are categories, functors, and natural transformations:

**Definition A.1** (Category). A *category* $\mathbf{C} = (\mathrm{Obj}, \mathrm{Hom}, \circ, \mathrm{id})$ consists of

- a collection $\mathrm{Obj}$ of *objects*,
- a set $\mathrm{Hom}(A, B)$ of *morphisms* between objects,
- an *identity morphism* $\mathrm{id}_A \in \mathrm{Hom}(A, A)$ for each object, and
- a *composition function* $\circ : \mathrm{Hom}(B, C) \times \mathrm{Hom}(A, B) \to \mathrm{Hom}(A, C)$ for each triplet of objects,

subject to *identity* $\mathrm{id}_B \circ f = f = f \circ \mathrm{id}_A$ and *associativity* $(h \circ g) \circ f = h \circ (g \circ f)$.

**Definition A.2** (Functor). For categories $\mathbf{C}$ and $\mathbf{D}$, a *functor* $F : \mathbf{C} \to \mathbf{D}$ consists of two mappings, which send

- $\mathbf{C}$-objects to $\mathbf{D}$-objects $F : \mathrm{Obj}_{\mathbf{C}} \to \mathrm{Obj}_{\mathbf{D}}$ and
- $\mathbf{C}$-morphisms to $\mathbf{D}$-morphisms $F : \mathrm{Hom}_{\mathbf{C}}(A, B) \to \mathrm{Hom}_{\mathbf{D}}(FA, FB)$

such that $F$ preserves *identity* $F\mathrm{id}_A = \mathrm{id}_{FA}$ and *composition* $F(g \circ_{\mathbf{C}} f) = Fg \circ_{\mathbf{D}} Ff$.

**Definition A.3** (Natural transformation). A *natural transformation* $\alpha : F \Rightarrow G$ between two functors $F, G : \mathbf{C} \to \mathbf{D}$ is a collection of $\mathbf{D}$-morphisms $\alpha_A : FA \to GA$ indexed by $\mathbf{C}$-objects, satisfying the *naturality* condition

$$
\begin{array}{ccc}
FA & \xrightarrow{Ff} & FB \\
\alpha_A \downarrow & & \downarrow \alpha_B \\
GA & \xrightarrow{Gf} & GB
\end{array}
\tag{26a}
$$

$$
\alpha_B \circ Ff = Gf \circ \alpha_A.
\tag{26b}
$$

In this work, we will work with the following categorical ingredients:

- a category $\mathbf{C}$ with finite products
- an endofunctor $F : \mathbf{C} \to \mathbf{C}$
- $\mathbf{C}$-objects $V$ and $V'$
- an arity $n \in \mathbb{N}$

## A.2. Algebra and coalgebra

Next, we review the definitions of algebras and coalgebras for an endofunctor and present a key result that induces functors between coalgebra categories.

**Definition A.4** (Algebra). An *F-algebra* $(A, a)$ is a $\mathbf{C}$-object $A$ together with a $\mathbf{C}$-morphism $a : FA \to A$.

**Definition A.5** (Coalgebra). An *F-coalgebra* $(A, a)$ is a $\mathbf{C}$-object $A$ together with a $\mathbf{C}$-morphism $a : A \to FA$.

**Definition A.6** (Algebra homomorphism). Given two $F$-algebras $(A, a)$ and $(B, b)$, an *F-algebra homomorphism* $f : (A, a) \to (B, b)$ is a $\mathbf{C}$-morphism $f : A \to B$ such that

$$
\begin{array}{ccc}
FA & \xrightarrow{Ff} & FB \\
a \downarrow & & \downarrow b \\
A & \xrightarrow{f} & B
\end{array}
\tag{27a}
$$

$$
f \circ a = b \circ Ff.
\tag{27b}
$$

The *category* $\mathbf{C}^F$ *of F-algebras* has $F$-algebras as objects and $F$-algebra homomorphisms as morphisms.

**Definition A.7** (Coalgebra homomorphism). Given two $F$-coalgebras $(A, a)$ and $(B, b)$, an $F$-*coalgebra homomorphism* $f : (A, a) \to (B, b)$ is a **C**-morphism $f : A \to B$ such that

$$
\begin{array}{ccc}
FA & \xrightarrow{Ff} & FB \\
a\uparrow & & \uparrow b \\
A & \xrightarrow{\phantom{Ff}f\phantom{Ff}} & B
\end{array}
\tag{28a}
$$

$$
Ff \circ a = b \circ f.
\tag{28b}
$$

The *category* $\mathbf{C}_F$ *of* $F$-*coalgebras* has $F$-coalgebras as objects and $F$-coalgebra homomorphisms as morphisms.

**Definition A.8** (Coalgebra-algebra homomorphism). Given an $F$-coalgebra $(A, a)$ and an $F$-algebra $(B, b)$, an $F$-*coalgebra-algebra homomorphism* $f : (A, a) \to (B, b)$ is a **C**-morphism $f : A \to B$ such that

$$
\begin{array}{ccc}
FA & \xrightarrow{Ff} & FB \\
a\uparrow & & \downarrow b \\
A & \xrightarrow{\phantom{Ff}f\phantom{Ff}} & B
\end{array}
\tag{29a}
$$

$$
Ff \circ a = b \circ f.
\tag{29b}
$$

Readers are referred to Rutten (2000); Jacobs (2016) for a comprehensive treatment of coalgebras and their applications.

In the main text, we primarily focus on the stochastic Moore machine functor $(\mathbb{P}(-))^A \times O$ with stochastic transition and deterministic observation. Other related functors modeling different types of systems include:

- Moore machine $(-)^A \times O$ with deterministic transition and observation,
- Moore machine $(-)^A \times O + 1$ with deterministic transition, observation, and termination,
- Moore machine $(\mathbb{P}(-))^A \times \mathbb{P}O$ with stochastic but independent transition and observation,
- Moore machine $\mathbb{P}((-)^A \times O)$ with correlated transition and observation, and
- Mealy machine $(\mathbb{P}(- \times O))^A$ with stochastic transition and action-dependent observation.

They capture different input-output interfaces and dependency structures.

## A.3. Hom-set and hom-functor

The *hom-set* $\mathrm{Hom}_{\mathbf{C}}(A, B)$ of a category $\mathbf{C}$ is the set of morphisms from $A$ to $B$. In particular, we focus on the hom-set $\mathrm{Hom}_{\mathbf{C}}(A, V)$ of morphisms from a **C**-object $A$ to a value object $V$. For each **C**-object $A$, we define a pointwise preorder $\preceq$ on the hom-set $\mathrm{Hom}_{\mathbf{C}}(A, V)$.

**Definition A.9** (Pointwise preorder). Let $\preceq_V : V \times V \to \{\top, \bot\}$ be an internal preorder on $V$. Write $\top_A : A \to \{\top, \bot\}$ for the constantly-true predicate on any **C**-object $A$. Define the pointwise preorder on $\mathrm{Hom}_{\mathbf{C}}(A, V)$ by

$$
\forall h, h' : A \to V.\ h \preceq h' := (\preceq_V \circ \langle h, h' \rangle = \top_A).
\tag{30}
$$

This is equivalent to the usual generalized-element formulation when **C** is well-pointed:

$$
\forall h, h' : A \to V.\ h \preceq h' := \forall a : 1 \to A.\ h \circ a \preceq h' \circ a.
\tag{31}
$$

**Definition A.10** (Order-preserving and order-reflecting function). A function $f : A \to B$ between preordered sets $(A, \preceq_A)$ and $(B, \preceq_B)$ is *order-preserving* if

$$
\forall a, a' \in A.\ (a \preceq_A a') \to (f(a) \preceq_B f(a')),
\tag{32}
$$

and *order-reflecting* if

$$
\forall a, a' \in A.\ (a \preceq_A a') \leftarrow (f(a) \preceq_B f(a')).
\tag{33}
$$

A key tool that we will use is the *Yoneda lemma*, which relates natural transformations from hom-functors to elements of functor values.

**Definition A.11** (Hom-functor). The *contravariant hom-functor* $\mathrm{Hom}_{\mathbf{C}}(-, V) : \mathbf{C}^{\mathrm{op}} \to \mathbf{Set}$, abbreviated as $H_V$,

$$H_V := \mathrm{Hom}_{\mathbf{C}}(-, V) : \quad \begin{array}{ccc} \mathbf{C}^{\mathrm{op}} & \to & \mathbf{Set} \\ B & \mapsto & \mathrm{Hom}_{\mathbf{C}}(B, V) \\ f : A \to B & \mapsto & \mathrm{Hom}_{\mathbf{C}}(f, V) : \mathrm{Hom}_{\mathbf{C}}(B, V) \to \mathrm{Hom}_{\mathbf{C}}(A, V) \end{array} \tag{34}$$

maps each $\mathbf{C}$-object $B$ to the hom-set of $\mathbf{C}$-morphisms from $B$ to $V$, and each $\mathbf{C}$-morphism $f : A \to B$ to the function $(-) \circ f$ that precomposes with $f$.

We denote $H_V^n := H_V \circ (-)^n = \mathrm{Hom}_{\mathbf{C}}((-)^n, V) : \mathbf{C}^{\mathrm{op}} \to \mathbf{Set}$ as the composition of the contravariant hom-functor $H_V$ and the product functor $(-)^n : \mathbf{C} \to \mathbf{C}$.

**Lemma A.12** (Yoneda lemma). *Given a contravariant functor $H : \mathbf{C}^{\mathrm{op}} \to \mathbf{Set}$, there exists a bijection between natural transformations $H_V \Rightarrow H$ and elements of $HV$:*

$$\mathrm{Hom}_{[\mathbf{C}^{\mathrm{op}}, \mathbf{Set}]}(H_V, H) \cong HV. \tag{35}$$

**Corollary A.13.** *Given two $\mathbf{C}$-objects $V$ and $V'$, there exists a bijection between natural transformations $H_V \Rightarrow H_{V'}$ and $\mathbf{C}$-morphisms $V \to V'$:*

$$\mathrm{Hom}_{[\mathbf{C}^{\mathrm{op}}, \mathbf{Set}]}(H_V, H_{V'}) \cong \mathrm{Hom}_{\mathbf{C}}(V, V'). \tag{36}$$

**Corollary A.14.** *Given an endofunctor $F : \mathbf{C} \to \mathbf{C}$, there exists a bijection between natural transformations $H_V \Rightarrow H_V F$ and $\mathbf{C}$-morphisms $FV \to V$:*

$$\mathrm{Hom}_{[\mathbf{C}^{\mathrm{op}}, \mathbf{Set}]}(H_V, H_V F) \cong \mathrm{Hom}_{\mathbf{C}}(FV, V). \tag{37}$$

Note that a $\mathbf{C}$-morphism $FV \to V$ is exactly an $F$-*algebra* on $V$.

# B. Lifting

Next, we introduce the general notion of functor lifting, which is widely used in coalgebraic modal logic (Kurz, 2001, Chapter 5); Pattinson (2003); Hasuo et al. (2007); Baldan et al. (2014). Special instances of functor lifting, such as relation lifting and metric lifting, have been used in the study of bisimulation (Kurz & Velebil, 2016).

The most general definition of functor lifting is as follows:

**Definition B.1** (Functor lifting). A *lifting* of an endofunctor $F : \mathbf{C} \to \mathbf{C}$ along a forgetful functor $U : \mathbf{D} \to \mathbf{C}$ is a functor $\mathbf{D}(F) : \mathbf{D} \to \mathbf{D}$ such that

$$
\begin{array}{ccc}
\mathbf{D} & \xrightarrow{\mathbf{D}(F)} & \mathbf{D} \\
U \downarrow & & \downarrow U \\
\mathbf{C} & \xrightarrow{F} & \mathbf{C}
\end{array}
\tag{38a}
$$

$$
U\mathbf{D}(F) = FU. \tag{38b}
$$

In other words, a lifted functor is the same functor but with a systematically chosen structure.

## B.1. Bundle lifting

Next, we introduce an instance of functor lifting, which we refer to as *bundle lifting*. We follow the following procedure:

**1. Define a category of bundles** A *bundle* over a $\mathbf{C}$-object $V$ is simply a $\mathbf{C}$-object $A$ equipped with a $\mathbf{C}$-morphism $h_A : A \to V$. A $n$-ary bundle $A^n \to V$ is a generalization where the domain is the $n$-fold product $A^n$.

**Definition B.2** (Bundle). A *$n$-ary bundle* over $V$ is a pair $(A, h_A : A^n \to V)$ of a $\mathbf{C}$-object $A$ and a $\mathbf{C}$-morphism $h_A : A^n \to V$ from the product $A^n$ to $V$. A *lax bundle morphism* $f : (A, h_A) \to (B, h_B)$ between $n$-ary bundles is a $\mathbf{C}$-morphism $f : A \to B$ such that

$$
\begin{array}{ccc}
A^n & \xrightarrow{f^n} & B^n \\
& \overset{\preceq}{\underset{h_A \searrow \swarrow h_B}{}} & \\
& V &
\end{array}
\tag{39a}
$$

$$
h_A \preceq h_B \circ f^n. \tag{39b}
$$

The category $\mathbf{C}_V^n$ of $n$-ary bundles over $V$ has $n$-ary bundles as objects and lax bundle morphisms as morphisms.

When the preorder $\preceq$ is the equality, $\mathbf{C}_V^n$ is the comma category $(-)^n \downarrow V$. When $n = 1$, $\mathbf{C}_V^1$ is the slice category $\mathbf{C} \downarrow V$.

**2. Define a forgetful functor**

**Definition B.3.** A forgetful functor $U : \mathbf{C}_V^n \to \mathbf{C}$ is given by

$$
U : \quad
\begin{array}{ccc}
\mathbf{C}_V^n & \to & \mathbf{C} \\
(A, h_A : A^n \to V) & \mapsto & A \\
f : (A, h_A) \to (B, h_B) & \mapsto & f : A \to B
\end{array}
\tag{40}
$$

**3. Define a functor lifting**

**Definition B.4** (Bundle lifting). The lifting $\mathbf{C}_V^n(F) : \mathbf{C}_V^n \to \mathbf{C}_V^n$ of an endofunctor $F : \mathbf{C} \to \mathbf{C}$ along $U : \mathbf{C}_V^n \to \mathbf{C}$ must have the form

$$
\mathbf{C}_V^n(F) : \quad
\begin{array}{ccc}
\mathbf{C}_V^n & \to & \mathbf{C}_V^n \\
(A, h_A : A^n \to V) & \mapsto & (FA, \lambda_A(h_A) : (FA)^n \to V) \\
f : (A, h_A) \to (B, h_B) & \mapsto & Ff : (FA, \lambda_A(h_A)) \to (FB, \lambda_B(h_B))
\end{array}
\tag{41}
$$

where $\lambda_A$ is a family of functions indexed by $\mathbf{C}$-objects $A$, mapping each $\mathbf{C}$-morphism $h_A : A^n \to V$ to a $\mathbf{C}$-morphism

$\lambda_A(h_A) : (FA)^n \to V$ such that

$$
\begin{array}{ccc}
A^n & \xrightarrow{\ f^n\ } & B^n \\
\end{array}
$$

(42a)

$$\forall h_A : A^n \to V. \ \forall h_B : B^n \to V. \ \forall f : A \to B. \ (h_A \preceq h_B \circ f^n) \to (\lambda_A(h_A) \preceq \lambda_B(h_B) \circ (Ff)^n). \tag{42b}$$

The following two properties of bundle liftings are immediate consequences of Def. B.4.

**A lifting corresponds to a modality**  A *modality* is an (oplax) natural transformation between contravariant hom-functors (Kurz, 2001). We show that a bundle lifting corresponds to a modality.

**Proposition B.5.** *The family of functions $\lambda_A : H_V^n A \to H_V^n F A$ of a lifting in Def. B.4 forms an oplax natural transformation $\lambda : H_V^n \Rightarrow H_V^n F$ between contravariant functors, called an $n$-ary modality:*

$$
\begin{array}{ccc}
H_V^n A & \xleftarrow{\ H_V^n f\ } & H_V^n B \\
\lambda_A \downarrow & \preceq & \downarrow \lambda_B \\
H_V^n F A & \xleftarrow[H_V^n F f]{} & H_V^n F B
\end{array}
\tag{43a}
$$

$$\lambda_A \circ H_V^n f \preceq H_V^n F f \circ \lambda_B. \tag{43b}$$

*Proof.* The functor lifting condition of $\mathbf{C}_V^n(F)$ in Eq. (42b) implies the oplax naturality condition of $\lambda$ in Eq. (43b) by taking $h_A = h_B \circ f^n$. $\qquad\square$

The following proposition shows that an order-preserving modality defines a functor lifting.

**Proposition B.6.** *If an $n$-ary modality $\lambda : H_V^n \Rightarrow H_V^n F$ is order-preserving, i.e., $(h \preceq h') \to (\lambda_A(h) \preceq \lambda_A(h'))$, then it defines a functor lifting $\mathbf{C}_V^n(F) : \mathbf{C}_V^n \to \mathbf{C}_V^n$ of $F$.*

*Proof.* If $h_A \preceq h_B \circ f^n$, then by order preservation, $\lambda_A(h_A) \preceq \lambda_A(h_B \circ f^n)$. By oplax naturality, $\lambda_A(h_B \circ f^n) \preceq \lambda_B(h_B) \circ (Ff)^n$. By transitivity, $\lambda_A(h_A) \preceq \lambda_B(h_B) \circ (Ff)^n$. $\qquad\square$

**A unary modality is an $F$-algebra**  According to Corollary A.14, when $n = 1$, there exists a bijection between modalities $\lambda : H_V \Rightarrow H_V F$ and $F$-algebras $f_V : FV \to V$ such that a bundle $A \xrightarrow{h_A} V$ is lifted to $FA \xrightarrow{Fh_A} FV \xrightarrow{f_V} V$.

When $n > 1$, however, not every modality comes from a $\mathbf{C}$-morphism $f_V : (FV)^n \to V$, as shown in the following counterexample:

**Example B.7.** For $n = 2$ and $F = \mathrm{id}_{\mathbf{C}}$, we have the following counterexample: $\lambda_A : H_V^2 A \to H_V^2 A : r_A \mapsto r_A \circ \langle p_2, p_1 \rangle$, where $p_1, p_2 : A \times A \to A$ are the projections. $\lambda$ is natural in $A$ but cannot be represented as $f_V \circ (-)$ for any $f_V : V \to V$.

### B.2. Unary bundle lifting

Next, we investigate bundle liftings for $n = 1$. We consider the category $\mathbf{P} := \mathbf{Set}_{\{\top, \bot\}}^1 = \mathbf{Set} \downarrow \{\top, \bot\}$ of predicates and the category $\mathbf{Q} := \mathbf{Set}_{[0,\infty]}^1 = \mathbf{Set} \downarrow [0,\infty]$ of quantities. We generalize the lifting in Jacobs (2016, Definition 6.1.1) by instantiating Lemma A.12 for polynomial functors and using generic operators.

**Constant functor**  The constant functor $\Delta X : \mathbf{C} \to \mathbf{C}$ maps every $\mathbf{C}$-object to a fixed $\mathbf{C}$-object $X$ and every $\mathbf{C}$-morphism to the identity morphism $\mathrm{id}_X$.

**Proposition B.8.** *Given two $\mathbf{C}$-objects $V$ and $X$, the following bijection holds:*

$$\mathrm{Hom}_{[\mathbf{C}^{\mathrm{op}}, \mathbf{Set}]}(H_V, H_V \Delta X) \cong \mathrm{Hom}_{\mathbf{C}}(X, V). \tag{44}$$

A lifting of the constant functor is just picking a fixed morphism $\lambda_A^{\Delta X}(h_A) : X \to V$.

**Identity functor**   The identity functor $\mathrm{id}_{\mathbf{C}} : \mathbf{C} \to \mathbf{C}$ maps every $\mathbf{C}$-object and every $\mathbf{C}$-morphism to itself.

**Proposition B.9.** *The following bijection holds:*

$$\mathrm{Hom}_{[\mathbf{C}^{\mathrm{op}}, \mathbf{Set}]}(H_V, H_V) \cong \mathrm{Hom}_{\mathbf{C}}(V, V). \tag{45}$$

A lifting of the identity functor is just postcomposing an endomorphism $\lambda_A^{\mathrm{id}_{\mathbf{C}}}(h_A) : A \xrightarrow{h_A} V \to V$.

**Exponentiation functor**   The functor $F^X : \mathbf{C} \to \mathbf{C}$ is the composition of an endofunctor $F : \mathbf{C} \to \mathbf{C}$ and the exponential functor $(-)^X : \mathbf{C} \to \mathbf{C}$, which maps every $\mathbf{C}$-object $A$ to the exponential object $A^X$ and every $\mathbf{C}$-morphism $f : A \to B$ to the exponential morphism $f^X : A^X \to B^X = f \circ (-)$.

**Proposition B.10.** *If $\mathbf{C}$ has exponentials, given an $\mathbf{C}$-object $X$, the following bijection holds:*

$$\mathrm{Hom}_{[\mathbf{C}^{\mathrm{op}}, \mathbf{Set}]}\left(H_V, H_V F^X\right) \cong \mathrm{Hom}_{\mathbf{C}}((FV)^X, V). \tag{46}$$

Note that $\mathrm{Hom}_{\mathbf{C}}((FV)^X, V)$ is richer than $\mathrm{Hom}_{\mathbf{C}}(FV, V)$. A construction of a lifting is as follows:

**Definition B.11.** Let $\square_X : V^X \to V$ be an *aggregator*. We can construct a lifting using $\square_X$ as follows:

$$
\begin{array}{ccccc}
\mathrm{Hom}_{\mathbf{C}}(FV, V) & \times & \mathrm{Hom}_{\mathbf{C}}(V^X, V) & \to & \mathrm{Hom}_{\mathbf{C}}((FV)^X, V) \\
f_V & & \square_X & \mapsto & \square_X \circ f_V^X
\end{array}
\tag{47}
$$

**Product functor**   The functor $F_1 \times F_2 : \mathbf{C} \to \mathbf{C}$ is the pointwise product of two endofunctors $F_1, F_2 : \mathbf{C} \to \mathbf{C}$, mapping a $\mathbf{C}$-object $A$ to $F_1 A \times F_2 A$ and a $\mathbf{C}$-morphism $f : A \to B$ to $F_1 f \times F_2 f : F_1 A \times F_2 A \to F_1 B \times F_2 B$.

**Proposition B.12.** *Given two endofunctors $F_1, F_2 : \mathbf{C} \to \mathbf{C}$, the following bijection holds:*

$$\mathrm{Hom}_{[\mathbf{C}^{\mathrm{op}}, \mathbf{Set}]}(H_V, H_V(F_1 \times F_2)) \cong \mathrm{Hom}_{\mathbf{C}}(F_1 C \times F_2 C, V). \tag{48}$$

Similarly to the exponentiation functor, a construction of a lifting is as follows:

**Definition B.13.** Let $\oplus : V \times V \to V$ be a *combinator*. We can construct a lifting using $\oplus$ as follows:

$$
\begin{array}{cccccc}
\mathrm{Hom}_{\mathbf{C}}(F_1 C, V) & \times & \mathrm{Hom}_{\mathbf{C}}(F_2 C, V) & \times \mathrm{Hom}_{\mathbf{C}}(V \times V, V) & \to & \mathrm{Hom}_{\mathbf{C}}(F_1 C \times F_2 C, V) \\
c_1 & & c_2 & \oplus & \mapsto & \oplus \circ (c_1 \times c_2)
\end{array}
\tag{49}
$$

**Coproduct functor**   The functor $F_1 + F_2 : \mathbf{C} \to \mathbf{C}$ is the pointwise coproduct of two endofunctors $F_1, F_2 : \mathbf{C} \to \mathbf{C}$, mapping a $\mathbf{C}$-object $A$ to $F_1 A + F_2 A$ and a $\mathbf{C}$-morphism $f : A \to B$ to $F_1 f + F_2 f : F_1 A + F_2 A \to F_1 B + F_2 B$.

**Proposition B.14.** *If $\mathbf{C}$ has finite coproducts, given two endofunctors $F_1, F_2 : \mathbf{C} \to \mathbf{C}$, the following bijection holds:*

$$\mathrm{Hom}_{[\mathbf{C}^{\mathrm{op}}, \mathbf{Set}]}(H_V, H_V(F_1 + F_2)) \cong \mathrm{Hom}_{\mathbf{C}}(F_1 C + F_2 C, V) \tag{50}$$

$$\cong \mathrm{Hom}_{\mathbf{C}}(F_1 C, V) \times \mathrm{Hom}_{\mathbf{C}}(F_2 C, V). \tag{51}$$

Due to the universal property of coproducts, a lifting of the coproduct functor is just given by the liftings of its components.

**Probability functor**   Lastly, we assume that a probability functor $\mathbb{P}$ is well-defined on the category $\mathbf{C}$ (Fritz, 2020). The functor $\mathbb{P}F : \mathbf{C} \to \mathbf{C}$ is the composition of an endofunctor $F : \mathbf{C} \to \mathbf{C}$ and the probability functor $\mathbb{P} : \mathbf{C} \to \mathbf{C}$, which maps a set $A$ to the set $\mathbb{P}A$ of probability distributions over $A$ and a function $f : A \to B$ to the function $\mathbb{P}f : \mathbb{P}A \to \mathbb{P}B$ defined by pushforward of distributions.

**Proposition B.15.** *Given a probability functor $\mathbb{P} : \mathbf{C} \to \mathbf{C}$, the following bijection holds:*

$$\mathrm{Hom}_{[\mathbf{C}^{\mathrm{op}}, \mathbf{Set}]}(H_V, H_V \mathbb{P}F) \cong \mathrm{Hom}_{\mathbf{C}}(\mathbb{P}FV, V). \tag{52}$$

We can use an extra operator to construct a lifting as follows:

**Definition B.16.** Let $\odot : \mathbb{P}V \to V$ be a *barycenter operator*. We can construct a lifting using $\odot$ as follows:

$$
\begin{array}{ccccc}
\mathrm{Hom}_{\mathbf{C}}(FV, V) & \times & \mathrm{Hom}_{\mathbf{C}}(\mathbb{P}V, V) & \to & \mathrm{Hom}_{\mathbf{C}}(\mathbb{P}FV, V) \\
f_V & & \odot & \mapsto & \odot \circ \mathbb{P}f_V
\end{array}
\tag{53}
$$

### B.3. Binary bundle lifting

Next, we investigate bundle liftings for $n = 2$. We consider the category $\mathbf{R} := \mathbf{Set}^2_{\{\top,\bot\}}$ of relations and the category $\mathbf{D} := \mathbf{Set}^2_{[0,\infty]}$ of distances. We generalize the relation lifting in Jacobs (2016, Definition 3.1.1); Kurz & Velebil (2016) by allowing generic operators. Since the Yoneda lemma in Lemma A.12 does not directly apply to $n > 1$, we need to consider liftings as natural transformations $\lambda^F : H^n_V \Rightarrow H^n_V F$.

**Exponentiation functor** $\quad (F^X)^n \cong (F^n)^X$

**Definition B.17.** Let $\square_X : V^X \to V$ be an *aggregator*. By slight abuse of notation, we denote the natural transformation $\square_X : H_V \Rightarrow H_V(-)^X$ with the same symbol. Given $\lambda^F : H^n_V \Rightarrow H^n_V F$, we can construct a lifting using $\square_X$ as follows:

$$\lambda^{F^X} : H^n_V \xLongrightarrow{\lambda^F} H^n_V F \cong H_V F^n$$

$$\xLongrightarrow{\square_X F^n} H_V(F^n)^X \cong H_V(F^X)^n \cong H^n_V F^X. \tag{54}$$

In other words, $\lambda^{F^X}_A(h_A) : ((FA)^X)^n \cong ((FA)^n)^X \xrightarrow{\lambda^F_A(h_A)^X} V^X \xrightarrow{\square_X} V$.

**Product functor** $\quad (F_1 \times F_2)^n \cong F_1^n \times F_2^n$

**Definition B.18.** Let $\oplus : V \times V \to V$ be a *combinator*, and let $\mathrm{Hom}_{\mathbf{C}}(-, \oplus) : H_{V \times V} \Rightarrow H_V$ be the Yoneda embedding. Given $\lambda^{F_1} : H^n_V \Rightarrow H^n_V F_1$ and $\lambda^{F_2} : H^n_V \Rightarrow H^n_V F_2$, we can construct a lifting using $\oplus$ as follows:

$$\lambda^{F_1 \times F_2} : H^n_V \xLongrightarrow{\langle \lambda^{F_1}, \lambda^{F_2} \rangle} H^n_V F_1 \times H^n_V F_2 \cong H_{V \times V}(F_1^n \times F_2^n)$$

$$\xLongrightarrow{\mathrm{Hom}_{\mathbf{C}}(-, \oplus)(F_1^n \times F_2^n)} H_V(F_1^n \times F_2^n) \cong H_V(F_1 \times F_2)^n \cong H^n_V(F_1 \times F_2). \tag{55}$$

In other words, $\lambda^{F_1 \times F_2}_A(h_A) : (F_1 A \times F_2 A)^n \cong (F_1 A)^n \times (F_2 A)^n \xrightarrow{\lambda^{F_1}_A(h_A) \times \lambda^{F_2}_A(h_A)} V \times V \xrightarrow{\oplus} V$.

### B.4. Probability lifting

For the probability functor, we present two constructions of liftings $\lambda^{\mathbb{P}F} : H^n_V \Rightarrow H^n_V \mathbb{P}F$: independent coupling and Wasserstein lifting.

**Independent coupling** $\quad$ Note that $\mathbb{P}$ is a lax monoidal functor with $\nabla : \mathbb{P}^n \Rightarrow \mathbb{P}(-)^n$ (Fritz, 2020, Proposition 3.1). A simple construction of a lifting is as follows:

**Definition B.19.** Let $\odot : \mathbb{P}V \to V$ be a *barycenter operator*. By slight abuse of notation, we denote the natural transformation $\odot : H_V \Rightarrow H_V \mathbb{P}$ with the same symbol. Given a lifting $\lambda^F : H^n_V \Rightarrow H^n_V F$, we can construct a lifting using $\odot$ as follows:

$$\lambda^{\mathbb{P}F} : H^n_V \xLongrightarrow{\lambda^F} H^n_V F \cong H_V F^n$$

$$\xLongrightarrow{\odot F^n} H_V \mathbb{P}F^n \tag{56}$$

$$\xLongrightarrow{H_V \nabla F} H_V(\mathbb{P}F)^n \cong H^n_V \mathbb{P}F.$$

In other words, $\lambda^{\mathbb{P}F}_A(h_A) : (\mathbb{P}FA)^n \xrightarrow{\nabla_{FA}} \mathbb{P}(FA)^n \xrightarrow{\mathbb{P}\lambda^F_A(h_A)} \mathbb{P}V \xrightarrow{\odot} V$.

**Wasserstein lifting**    We first introduce the notion of coupling.

**Definition B.20** (Coupling fiber). Let $p_1, p_2 : A \times A \to A$ be the projections, and let $\Delta_A : \mathbb{P}(A \times A) \to \mathbb{P}A \times \mathbb{P}A :=$ $\langle \mathbb{P}p_1, \mathbb{P}p_2 \rangle$ be the marginalization map. A *coupling* of two probability distributions $\mu, \nu \in \mathbb{P}A$ is a probability distribution $\pi_A \in \mathbb{P}(A \times A)$ such that $\Delta_A(\pi_A) = \langle \mu, \nu \rangle$. Equivalently, the set $\Pi_A(\mu, \nu)$ of all couplings of $\mu$ and $\nu$ is given by the following pullback:

$$
\begin{array}{ccc}
\Pi_A(\mu, \nu) & \longrightarrow & 1 \\
\downarrow & \lrcorner & \downarrow {\scriptstyle \langle \mu, \nu \rangle} \\
\mathbb{P}(A \times A) & \xrightarrow{\Delta_A} & \mathbb{P}A \times \mathbb{P}A
\end{array}
\tag{57}
$$

**Lemma B.21.** *Given a map $f : A \to B$ and two probability distributions $\mu, \nu \in \mathbb{P}A$ over $A$, if $\pi_A \in \mathbb{P}(A \times A)$ is a coupling of $\mu$ and $\nu$, then the pushforward distribution $(f \times f)_* \pi_A \in \mathbb{P}(B \times B)$ is a coupling of $f_* \mu$ and $f_* \nu$; if $\pi_B \in \mathbb{P}(B \times B)$ is a coupling of $f_* \mu$ and $f_* \nu$ and $f$ has a retraction $r : B \to A$, then the pushforward distribution $(r \times r)_* \pi_B \in \mathbb{P}(A \times A)$ is a coupling of $\mu$ and $\nu$.*

*Proof.* Note that the pushforward commutes with marginalization:

$$
(\mathbb{P}f \times \mathbb{P}f) \circ \Delta_A = \Delta_B \circ \mathbb{P}(f \times f).
\tag{58}
$$

Then, we have the following equality of compositions:

$$
\Pi_A(\mu, \nu) \rightarrowtail \mathbb{P}(A \times A) \xrightarrow{\mathbb{P}(f \times f)} \mathbb{P}(B \times B) \xrightarrow{\Delta_B} \mathbb{P}B \times \mathbb{P}B = \Pi_A(\mu, \nu) \to 1 \xrightarrow{\langle f_* \mu, f_* \nu \rangle} \mathbb{P}B \times \mathbb{P}B.
\tag{59}
$$

By the universal property, there is a unique morphism $\Pi_A(\mu, \nu) \to \Pi_B(f_* \mu, f_* \nu)$, as shown in the following diagram:

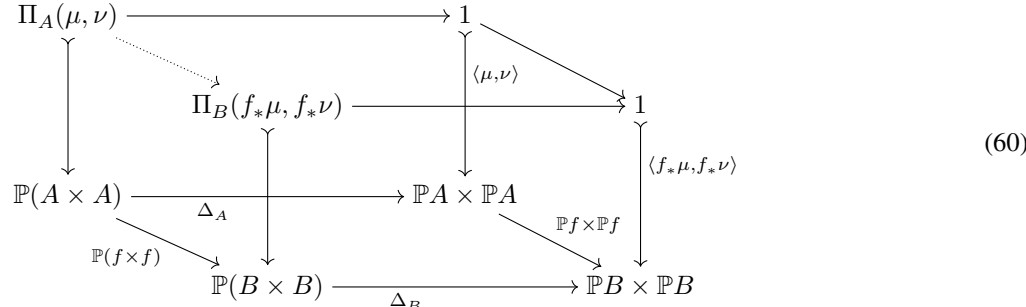

$$
\tag{60}
$$

which means that

$$
\{ (f \times f)_* \pi_A \mid \pi_A \in \Pi_A(\mu, \nu) \} \subseteq \Pi_B(f_* \mu, f_* \nu).
\tag{61}
$$

If $r : B \to A$ is a retraction of $f : A \to B$, i.e., $r \circ f = \mathrm{id}_A$, then

$$
(\mathbb{P}r \times \mathbb{P}r) \circ \langle f_* \mu, f_* \nu \rangle = \langle \mathbb{P}r \circ f_* \mu, \mathbb{P}r \circ f_* \nu \rangle = \langle (r \circ f)_* \mu, (r \circ f)_* \nu \rangle = \langle \mu, \nu \rangle.
\tag{62}
$$

$$
\mathbb{P}(r \times r) \circ \mathbb{P}(f \times f) = \mathbb{P}((r \times r) \circ (f \times f)) = \mathbb{P}(r \circ f) \times (r \circ f) = \mathbb{P}(\mathrm{id}_A \times \mathrm{id}_A) = \mathbb{P}\mathrm{id}_{A \times A} = \mathrm{id}_{\mathbb{P}(A \times A)}.
\tag{63}
$$

$$
(\mathbb{P}r \times \mathbb{P}r) \circ (\mathbb{P}f \times \mathbb{P}f) = (\mathbb{P}r \circ \mathbb{P}f) \times (\mathbb{P}r \circ \mathbb{P}f) = \mathbb{P}(r \circ f) \times \mathbb{P}(r \circ f) = \mathbb{P}\mathrm{id}_A \times \mathbb{P}\mathrm{id}_A = \mathrm{id}_{\mathbb{P}A} \times \mathrm{id}_{\mathbb{P}A} = \mathrm{id}_{\mathbb{P}A \times \mathbb{P}A}.
\tag{64}
$$

Then, there is a unique morphism $\Pi_B(f_* \mu, f_* \nu) \to \Pi_A(\mu, \nu)$ such that

$$
\{ (r \times r)_* \pi_B \mid \pi_B \in \Pi_B(f_* \mu, f_* \nu) \} \subseteq \Pi_A(\mu, \nu).
\tag{65}
$$

$\square$

**Definition B.22.** Let $\vee$ be the join operator in the preordered set $(V, \preceq)$. Given a **C**-object $A$, we define a function

$$
\begin{aligned}
\kappa_A : \quad \mathrm{Hom}_{\mathbf{C}}(\mathbb{P}(A \times A), V) \quad &\to \quad \mathrm{Hom}_{\mathbf{C}}(\mathbb{P}A \times \mathbb{P}A, V) \\
g \quad &\mapsto \quad \left[ (\mu, \nu) \mapsto \bigvee_{\pi \in \Pi_A(\mu, \nu)} g(\pi) \right].
\end{aligned}
\tag{66}
$$

**Proposition B.23.** *The family of functions $\kappa_A : \mathrm{Hom}_{\mathbf{C}}(\mathbb{P}(A \times A), V) \to \mathrm{Hom}_{\mathbf{C}}(\mathbb{P}A \times \mathbb{P}A, V)$ in Def. B.22 forms an oplax natural transformation $\kappa : H_V \mathbb{P}(-)^2 \Rightarrow H_V^2 \mathbb{P}$.*

*Proof.* That $\kappa$ is an oplax natural transformation means that for every **C**-morphism $f : A \to B$, the following diagram

commutes:

$$\begin{array}{ccc}
\mathrm{Hom}_{\mathbf{C}}(\mathbb{P}(A \times A), V) & \xleftarrow{\mathrm{Hom}_{\mathbf{C}}(\mathbb{P}(f \times f), V)} & \mathrm{Hom}_{\mathbf{C}}(\mathbb{P}(B \times B), V) \\
{\scriptstyle \kappa_A}\downarrow & \preceq & \downarrow{\scriptstyle \kappa_B} \\
\mathrm{Hom}_{\mathbf{C}}(\mathbb{P}A \times \mathbb{P}A, V) & \xleftarrow[\mathrm{Hom}_{\mathbf{C}}(\mathbb{P}f \times \mathbb{P}f, V)]{} & \mathrm{Hom}_{\mathbf{C}}(\mathbb{P}B \times \mathbb{P}B, V)
\end{array} \tag{67a}$$

$$\kappa_A \circ \mathrm{Hom}_{\mathbf{C}}(\mathbb{P}(f \times f), V) \preceq \mathrm{Hom}_{\mathbf{C}}(\mathbb{P}f \times \mathbb{P}f, V) \circ \kappa_B, \tag{67b}$$

$$\forall g : \mathbb{P}(B \times B) \to V.\ \kappa_A(g \circ \mathbb{P}(f \times f)) \preceq \kappa_B(g) \circ (\mathbb{P}f \times \mathbb{P}f), \tag{67c}$$

$$\forall g : \mathbb{P}(B \times B) \to V.\ \forall \mu \in \mathbb{P}A.\ \forall \nu \in \mathbb{P}A.\ \bigvee_{\pi_A \in \Pi_A(\mu,\nu)} g((f \times f)_* \pi_A) \preceq \bigvee_{\pi_B \in \Pi_B(f_*\mu, f_*\nu)} g(\pi_B), \tag{67d}$$

which is a result of Lemma B.21 and the monotonicity of $\vee$. $\qquad\square$

We can further show that $\kappa_A$ is a left adjoint of the precomposition of marginalization.

**Proposition B.24.** $\kappa_A$ *is a left adjoint of the precomposition* $\Delta_A{}^* : \mathrm{Hom}_{\mathbf{C}}(\mathbb{P}A \times \mathbb{P}A, V) \to \mathrm{Hom}_{\mathbf{C}}(\mathbb{P}(A \times A), V)$ *of marginalization* $\Delta_A : \mathbb{P}(A \times A) \to \mathbb{P}A \times \mathbb{P}A$:

$$\mathrm{Hom}_{\mathbf{C}}(\mathbb{P}(A \times A), V) \underset{\Delta_A{}^*}{\overset{\kappa_A}{\underset{\perp}{\rightleftarrows}}} \mathrm{Hom}_{\mathbf{C}}(\mathbb{P}A \times \mathbb{P}A, V), \tag{68}$$

*which satisfies the following adjunction property:*

$$\forall g : \mathbb{P}(A \times A) \to V.\ \forall h : \mathbb{P}A \times \mathbb{P}A \to V.\ (\kappa_A(g) \preceq h) \leftrightarrow (g \preceq h \circ \Delta_A). \tag{69}$$

*Proof.* Let $\mu, \nu \in \mathbb{P}A$ be probability distributions over $A$, and let $\pi \in \mathbb{P}(A \times A)$ be their coupling, i.e., $\Delta_A \circ \pi = \langle \mu, \nu \rangle$.

Assume $\kappa_A(g) \preceq h$. Then, $\pi$ factors through the fiber $\Pi_A(\mu, \nu)$, hence by definition of $\kappa_A(g)$ as a join over this fiber we have $g \circ \pi \preceq \kappa_A(g) \circ \langle \mu, \nu \rangle$. Composing the assumption with $\langle \mu, \nu \rangle$ yields $\kappa_A(g) \circ \langle \mu, \nu \rangle \preceq h \circ \langle \mu, \nu \rangle$, so we have $g \circ \pi \preceq h \circ \langle \mu, \nu \rangle = h \circ \Delta_A \circ \pi$. Since $\pi$ was arbitrary, this shows $g \preceq h \circ \Delta_A$.

Assume $g \preceq h \circ \Delta_A$. Composing the assumption with $\pi$ yields $g \circ \pi \preceq h \circ \Delta_A \circ \pi = h \circ \langle \mu, \nu \rangle$. Thus, $h \circ \langle \mu, \nu \rangle$ is an upper bound of the family $\{g \circ \pi \mid \pi \in \Pi_A(\mu, \nu)\}$, and since $\kappa_A(g) \circ \langle \mu, \nu \rangle$ is their join, we obtain $\kappa_A(g) \circ \langle \mu, \nu \rangle \preceq h \circ \langle \mu, \nu \rangle$. As $\langle \mu, \nu \rangle$ was arbitrary, $\kappa_A(g) \preceq h$ follows. $\qquad\square$

Finally, we can construct the Wasserstein lifting as follows:

**Definition B.25** (Wasserstein lifting). Let $\odot : \mathbb{P}V \to V$ be a *barycenter operator*. We apply $\kappa_{FA}$ in Def. B.22 to the composition $g_{h_A} : \mathbb{P}(FA \times FA) \xrightarrow{\mathbb{P}\lambda^F(h_A)} \mathbb{P}V \xrightarrow{\odot} V$. Given a lifting $\lambda^F : H_V^2 \Rightarrow H_V^2 F$, we can construct a lifting using $\odot$ and $\kappa$ as follows:

$$\begin{aligned}
\lambda^{\mathbb{P}F} : H_V^2 &\overset{\lambda^F}{\Longrightarrow} H_V^2 F \cong H_V F^2 \\
&\overset{\odot F^2}{\Longrightarrow} H_V \mathbb{P}F^2 \\
&\overset{\kappa F}{\Longrightarrow} H_V(\mathbb{P}F)^2 \cong H_V^2 \mathbb{P}F.
\end{aligned} \tag{70}$$

In other words, $\lambda_A^{\mathbb{P}F}(h_A) : \mathbb{P}FA \times \mathbb{P}FA \to V := \left[ (\mu, \nu) \mapsto \bigvee_{\pi \in \Pi_{FA}(\mu,\nu)} \odot \lambda^F(h_A)_* \pi \right]$.

Fig. 2 shows three different ways to lift a relation/distance $A \times A \to V$ to probability distributions $\mathbb{P}A \times \mathbb{P}A \to V$.

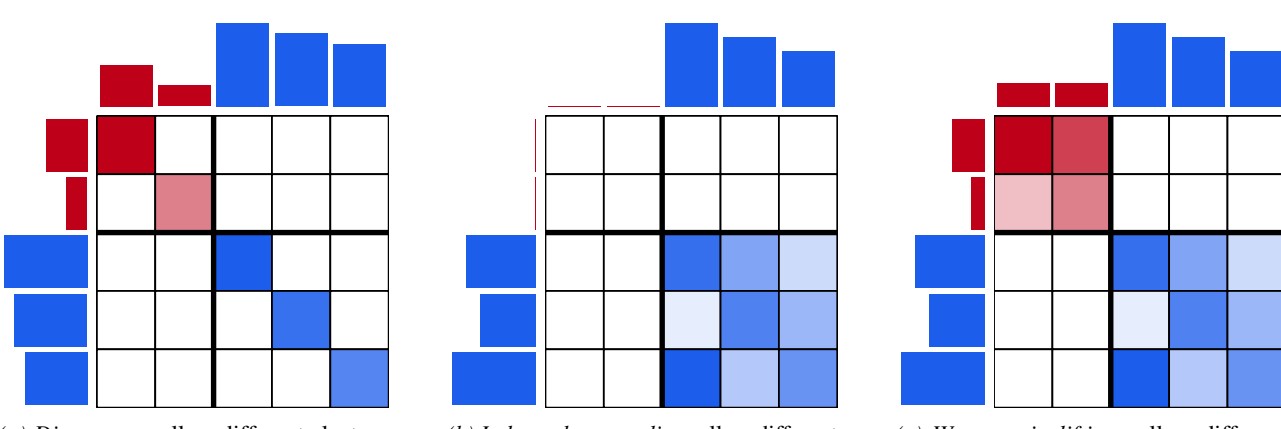

*(a) Divergence*: allow different clusters, disallow different instances with different probabilities

*(b) Independent coupling*: allow different instances within a single cluster, disallow different clusters

*(c) Wasserstein lifting*: allow different clusters and instances, disallow different total probabilities within clusters

*Figure 2.* Three ways to lift a relation/distance $A \times A \to V$ to probability distributions $\mathbb{P}A \times \mathbb{P}A \to V$.

### B.5. Commutative lifting

Next, we investigate the commutativity of liftings between different categories.

**Definition B.26** (Commutative functor lifting). Given two forgetful functors $U : \mathbf{D} \to \mathbf{C}$ and $U' : \mathbf{D}' \to \mathbf{C}$ and a functor $Z : \mathbf{D} \to \mathbf{D}'$ such that $U = U'Z$, the liftings $\mathbf{D}(F) : \mathbf{D} \to \mathbf{D}$ and $\mathbf{D}'(F) : \mathbf{D}' \to \mathbf{D}'$ of the same endofunctor $F : \mathbf{C} \to \mathbf{C}$ *commute* with $Z$ if

$$\begin{array}{ccc} \mathbf{D}' & \xrightarrow{\ \mathbf{D}'(F)\ } & \mathbf{D}' \\ & & \end{array} \tag{71a}$$

$$Z\mathbf{D}(F) = \mathbf{D}'(F)Z. \tag{71b}$$

For our bundle categories, the commutativity condition can be expressed at three different levels: functor level, natural transformation level, and morphism level.

**Functor level**   A $\mathbf{C}$-morphism $z : V \to V'$ induces a functor $\mathbf{C}_z^n : \mathbf{C}_V^n \to \mathbf{C}_{V'}^n$ between the corresponding bundle categories:

$$\begin{array}{cccc} \mathbf{C}_z^n : & \mathbf{C}_V^n & \to & \mathbf{C}_{V'}^n \\ & (A, h_A : A^n \to V) & \mapsto & (A, z \circ h_A : A^n \xrightarrow{h_A} V \xrightarrow{z} V') \\ & f : (A, h_A) \to (B, h_B) & \mapsto & f : (A, z \circ h_A) \to (B, z \circ h_B) \end{array} \tag{72}$$

Given two forgetful functors $U : \mathbf{C}_V^n \to \mathbf{C}$ and $U' : \mathbf{C}_{V'}^n \to \mathbf{C}$ defined in Def. B.3, we have $U = U'\mathbf{C}_z^n$. According to Def. B.26, two liftings $\mathbf{C}_V^n(F) : \mathbf{C}_V^n \to \mathbf{C}_V^n$ and $\mathbf{C}_{V'}^n(F) : \mathbf{C}_{V'}^n \to \mathbf{C}_{V'}^n$ of the same endofunctor $F : \mathbf{C} \to \mathbf{C}$ commute with $\mathbf{C}_z^n : \mathbf{C}_V^n \to \mathbf{C}_{V'}^n$ if

$$\mathbf{C}_z^n \mathbf{C}_V^n(F) = \mathbf{C}_{V'}^n(F)\mathbf{C}_z^n. \tag{73}$$

**Natural transformation level**   According to Corollary A.13, a $\mathbf{C}$-morphism $z : V \to V'$ corresponds to a natural transformation $\zeta : H_V \Rightarrow H_{V'}$. We denote $\zeta^n : H_V^n \Rightarrow H_{V'}^n := \zeta(-)^n$ as the vertical composition (whiskering) of the natural transformation $\zeta$ and the product functor $(-)^n$. Then, the commutativity condition can be expressed in terms of the

corresponding modalities $\lambda : H_V^n \Rightarrow H_V^n F$ and $\lambda' : H_{V'}^n \Rightarrow H_{V'}^n F$:

$$
\begin{array}{ccc}
H_V^n & \xRightarrow{\zeta^n} & H_{V'}^n \\
\lambda \Downarrow & & \Downarrow \lambda' \\
H_V^n F & \xRightarrow[\zeta^n F]{} & H_{V'}^n F
\end{array}
\tag{74a}
$$

$$
\zeta^n F \circ \lambda = \lambda' \circ \zeta^n,
\tag{74b}
$$

which means that $\lambda$ and $\lambda'$ are $F$-modules in the presheaf category $[\mathbf{C}^{\mathrm{OP}}, \mathbf{Set}]$ and $\zeta^n$ is an $F$-module homomorphism from $\lambda$ to $\lambda'$.

$$
\begin{array}{ccc}
H_V^n A & \xrightarrow{\zeta_A^n} & H_{V'}^n A \\
\lambda_A \downarrow & & \downarrow \lambda_A' \\
H_V^n F A & \xrightarrow[\zeta_{FA}^n]{} & H_{V'}^n F A
\end{array}
\tag{75a}
$$

$$
\zeta_{FA}^n \circ \lambda_A = \lambda_A' \circ \zeta_A^n.
\tag{75b}
$$

**Morphism level**  At the level of individual morphisms, the commutativity condition states that for every $n$-ary bundle $h_A : A^n \to V$, the following diagram commutes:

$$
\begin{array}{ccc}
 & A^n & \\
{\scriptstyle h_A} \swarrow & & \searrow {\scriptstyle z \circ h_A} \\
V & \xrightarrow{z} & V' \\
{\scriptstyle \lambda_A(h_A)} \searrow & & \nearrow {\scriptstyle \lambda_A'(z \circ h_A)} \\
 & (FA)^n &
\end{array}
\tag{76a}
$$

$$
\forall h_A : A^n \to V. \; z \circ \lambda_A(h_A) = \lambda_A'(z \circ h_A).
\tag{76b}
$$

For the unary case, the commutativity condition at the morphism level is equivalent to $z : V \to V'$ being an $F$-algebra homomorphism between the $F$-algebras defined by the liftings $\lambda$ and $\lambda'$.

**Proposition B.27.**  *When $n = 1$, the commutativity condition at the morphism level is equivalent to $z : V \to V'$ being an $F$-algebra homomorphism between the $F$-algebras defined by the liftings $\lambda$ and $\lambda'$.*

*Proof.* Let $h_A : A \to V$ be a unary bundle. We have $\lambda_A(h_A) = f_V \circ F h_A$ and $\lambda_A'(h_A) = f_{V'} \circ F h_A$ for some $F$-algebras $f_V : FV \to V$ and $f_{V'} : FV' \to V'$. Then, the commutativity condition in Eq. (76b) becomes

$$
z \circ \lambda_A(h_A) = \underline{z \circ f_V \circ F h_A = f_{V'} \circ F z \circ F h_A} = f_{V'} \circ F(z \circ h_A) = \lambda_A'(z \circ h_A).
\tag{77}
$$

By choosing $A = V$ and $h_A = \mathrm{id}_V$, this reduces to that $z : V \to V'$ is an $F$-algebra homomorphism $z : (V, f_V) \to (V', f_{V'})$:

$$
\begin{array}{ccc}
FV & \xrightarrow{Fz} & FV' \\
f_V \downarrow & & \downarrow f_{V'} \\
V & \xrightarrow{z} & V'
\end{array}
\tag{78a}
$$

$$
z \circ f_V = f_{V'} \circ Fz.
\tag{78b}
$$

$\square$

Such algebra homomorphism conditions have been used in Zhang & Sugiyama (2024) to characterize the relationship between logical definitions and quantitative metrics.

# C. Pullback

In this section, we detail the notion of pullback in the context of bundle lifting. Note that this is pullback in the sense of inverse image (opposite of pushforward or direct image), not in the sense of limit (opposite of pushout).

**Definition C.1** (Pullback morphism)**.** The *pullback* of a **C**-morphism $g : B \to V$ along a **C**-morphism $f : A \to B$ is a **C**-morphism $f^*g := g \circ f : A \to V$ defined by precomposition of $g$ with $f$.

Then, we can define the pullback of bundles as follows.

**Definition C.2** (Pullback bundle)**.** The pullback of an $n$-ary bundle $h_B : B^n \to V$ along a morphism $f : A \to B$ is the pullback morphism $(f^n)^* h_B := h_B \circ f^n$ along the product morphism $f^n : A^n \to B^n$. With slight abuse of notation, we may also write $f^* h_B$ for $(f^n)^* h_B$ when $n$ is clear from context.

$$
\begin{array}{ccc}
A^n & \xrightarrow{\;f^n\;} & B^n \\
 & \searrow^{\,f^* h_B} \quad \swarrow_{\,h_B} & \\
 & V &
\end{array}
\tag{79}
$$

In other words, the pullback is the contravariant functor $H_V^n : \mathbf{C}^{\mathrm{op}} \to \mathbf{Set}$ acting on morphisms:

$$
f^* := H_V^n f : \quad
\begin{aligned}
H_V^n B &\;\to\; H_V^n A \\
h_B &\;\mapsto\; h_B \circ f^n
\end{aligned}
\tag{80}
$$

Four special cases of pullback bundles are pullback predicates, quantities, relations, and distances.

**Definition C.3** (Pullback predicate)**.** Given a function $f : A \to B$ and a predicate $p : B \to \{\top, \bot\}$ on the codomain $B$, the *pullback predicate* $f^* p : A \to \{\top, \bot\} := p \circ f$ is a predicate on the domain $A$:

$$
\forall a \in A.\ f^* p(a) := p(f(a)). \tag{81}
$$

**Definition C.4** (Pullback quantity)**.** Given a function $f : A \to B$ and a quantity $q : B \to [0, \infty]$ on the codomain $B$, the *pullback quantity* $f^* q : A \to [0, \infty] := q \circ f$ is a quantity on the domain $A$:

$$
\forall a \in A.\ f^* q(a) := q(f(a)). \tag{82}
$$

**Definition C.5** (Pullback relation)**.** Given a function $f : A \to B$ and a relation $r : B \times B \to \{\top, \bot\}$ on the codomain $B$, the *pullback relation* $f^* r : A \times A \to \{\top, \bot\} := r \circ (f \times f)$ is a relation on the domain $A$:

$$
\forall a_1 \in A.\ \forall a_2 \in A.\ f^* r(a_1, a_2) := r(f(a_1), f(a_2)). \tag{83}
$$

**Definition C.6** (Pullback distance)**.** Given a function $f : A \to B$ and a distance $d : B \times B \to [0, \infty]$ on the codomain $B$, the *pullback distance* $f^* d : A \times A \to [0, \infty] := d \circ (f \times f)$ is a distance on the domain $A$:

$$
\forall a_1 \in A.\ \forall a_2 \in A.\ f^* d(a_1, a_2) := d(f(a_1), f(a_2)). \tag{84}
$$

In particular, the pullback of the equality relation $=_B : B \times B \to \{\top, \bot\}$ is called the *kernel* relation: $\ker f := f^*(=_B)$.

## C.1. Pullback of order

We can establish the order-preserving and order-reflecting properties of pullback bundles.

**Lemma C.7.** *The pullback $f^*$ is order-preserving.*

*Proof.* Given two bundles $h_B, h_B' : B^n \to V$, assume $h_B \preceq h_B'$. By the definition of the pointwise preorder in Def. A.9, $h_B \preceq h_B'$ means that $\preceq_V \circ \langle h_B, h_B' \rangle = \top_{B^n}$. Because $\langle f^* h_B, f^* h_B' \rangle = \langle h_B \circ f^n, h_B' \circ f^n \rangle = \langle h_B, h_B' \rangle \circ f^n$ and $\top_{A^n} = \top_{B^n} \circ f^n$, we have $\preceq_V \circ \langle h_B, h_B' \rangle \circ f^n = \top_{B^n} \circ f^n$. Hence, $\preceq_V \circ \langle f^* h_B, f^* h_B' \rangle = \top_{A^n}$, which is exactly $f^* h_B \preceq f^* h_B'$. $\qquad\square$

**Lemma C.8.** *If $f^n$ is an epimorphism, then the pullback $f^*$ is order-reflecting.*

*Proof.* Given two bundles $h_B, h_B' : B^n \to V$, assume $f^* h_B \preceq f^* h_B'$. By the definition of the pointwise preorder in Def. A.9, $f^* h_B \preceq f^* h_B'$ means that $\preceq_V \circ \langle h_B \circ f^n, h_B' \circ f^n \rangle = \top_{A^n}$. Because $\langle h_B \circ f^n, h_B' \circ f^n \rangle = \langle h_B, h_B' \rangle \circ f^n$ and $\top_{A^n} = \top_{B^n} \circ f^n$, we have $\preceq_V \circ \langle h_B, h_B' \rangle \circ f^n = \top_{B^n} \circ f^n$. Because $f^n$ is an epimorphism, precomposition with $f^n$ is cancellable, hence $\preceq_V \circ \langle h_B, h_B' \rangle = \top_{B^n}$, which is exactly $h_B \preceq h_B'$. $\qquad\square$

## C.2. Pullback of pullback

**Lemma C.9.** *Given an $F$-coalgebra homomorphism $f : (A, t_A) \to (B, t_B)$, we have*

$$
\begin{array}{ccc}
H_V^n FA & \xleftarrow{\ (Ff)^*\ } & H_V^n FB \\
{\scriptstyle t_A{}^*}\downarrow & & \downarrow{\scriptstyle t_B{}^*} \\
H_V^n A & \xleftarrow{\ f^*\ } & H_V^n B
\end{array}
\tag{85a}
$$

$$
t_A{}^* \circ (Ff)^* = f^* \circ t_B{}^*.
\tag{85b}
$$

$$
\forall h_B : B^n \to V.\ t_A{}^*(Ff)^* h_B = f^* t_B{}^* h_B.
\tag{85c}
$$

*Proof.* This is a direct result of the definition of pullback and the coalgebra homomorphism. ☐

## C.3. Pullback of lifting

**Lemma C.10.** *Given an $F$-coalgebra homomorphism $f : (A, t_A) \to (B, t_B)$ and a lifting $\lambda$, we have*

$$
\begin{array}{ccc}
H_V^n A & \xleftarrow{\ f^*\ } & H_V^n B \\
{\scriptstyle \lambda_A}\downarrow & \preceq & \downarrow{\scriptstyle \lambda_B} \\
H_V^n FA & \xleftarrow{\ (Ff)^*\ } & H_V^n FB
\end{array}
\tag{86a}
$$

$$
\lambda_A \circ f^* \preceq (Ff)^* \circ \lambda_B.
\tag{86b}
$$

$$
\forall h_B : B^n \to V.\ \lambda_A(f^* h_B) \preceq (Ff)^* \lambda_B(h_B).
\tag{86c}
$$

*If the lifting $\lambda$ is strict at $f$, then the above preorder is an equality.*

*Proof.* This is a direct result of the definition of lifting in Def. B.4, where $h_A = f^* h_B$. ☐

## C.4. Pullback of closure

The composition of a pullback and a lifting is of central importance in our development.

**Definition C.11** (Closure operator). Given a lifting $\lambda : H_V^n \Rightarrow H_V^n F$ and an $F$-coalgebra $t_A : A \to FA$, we define a *closure operator* $T_A : H_V^n A \to H_V^n A$ by

$$
T_A := t_A{}^* \circ \lambda_A : \quad
\begin{array}{ccccc}
H_V^n A & \xrightarrow{\ \lambda_A\ } & H_V^n FA & \xrightarrow{\ t_A{}^*\ } & H_V^n A \\
h_A & \mapsto & \lambda_A(h_A) & \mapsto & t_A{}^* \lambda_A(h_A)
\end{array}
\tag{87}
$$

A post-fixed point $h_A : A^n \to V$ of the closure operator $T_A : H_V^n A \to H_V^n A$ satisfies

$$
\begin{array}{ccc}
A^n & \xrightarrow{\ t_A^n\ } & (FA)^n \\
{\scriptstyle h_A}\searrow & \preceq & \swarrow{\scriptstyle \lambda_A(h_A)} \\
& V &
\end{array}
\tag{88a}
$$

$$
h_A \preceq T_A(h_A).
\tag{88b}
$$

For $n = 1$, such post-fixed points correspond to lax homomorphisms from the $F$-coalgebra to the $F$-algebra defined by the lifting.

**Proposition C.12.** *If $h_A : A \to V$ is a post-fixed point of a unary closure operator $T_A : H_V A \to H_V A$, then it is a lax homomorphism from the $F$-coalgebra $(A, t_A)$ to the $F$-algebra $(V, f_V)$ defined by the lifting $\lambda$:*

$$
\begin{array}{ccc}
A & \xrightarrow{\ t_A\ } & FA \\
{\scriptstyle h_A}\downarrow & \preceq & \downarrow{\scriptstyle Fh_A} \\
V & \xleftarrow{\ f_V\ } & FV
\end{array}
\tag{89a}
$$

$$
h_A \preceq f_V \circ Fh_A \circ t_A.
\tag{89b}
$$

To relate the closure operators on different coalgebras, we introduce the notion of laxity and oplaxity along pullback.

**Definition C.13** (Lax and oplax along pullback). Given $F$-coalgebras $(A, t_A)$ and $(B, t_B)$, a morphism $f : A \to B$, a lifting $\lambda$ of $n$-ary bundles, and the closure operators $T_A$ and $T_B$ induced by the lifting and the coalgebras, we say that the closure operators are *oplax* (resp. *lax*) *along pullback* $f^*$ if the following diagram commutes up to $\preceq$ (resp. $\succeq$):

$$
\begin{array}{ccc}
H_V^n A & \xleftarrow{\quad f^* \quad} & H_V^n B \\
\lambda_A \downarrow & & \downarrow \lambda_B \\
H_V^n FA & \preceq & H_V^n FB \\
t_A{}^* \downarrow & & \downarrow t_B{}^* \\
H_V^n A & \xleftarrow{\quad f^* \quad} & H_V^n B
\end{array}
\tag{90a}
$$

$$T_A \circ f^* \preceq f^* \circ T_B. \tag{90b}$$

$$\forall h_B : B^n \to V.\ T_A(f^* h_B) \preceq f^* T_B(h_B). \tag{90c}$$

$$\forall h_B : B^n \to V.\ \forall a_1, \ldots, a_n \in A.\ \lambda_A(f^* h_B)(t_A(a_1), \ldots, t_A(a_n)) \preceq \lambda_B(h_B)(t_B(f(a_1)), \ldots, t_B(f(a_n))). \tag{90d}$$

Then, we can establish the following two results about the preservation and reflection of post-fixed points along pullback.

**Lemma C.14.** *If the closure operators are lax along an order-preserving pullback $f^*$, then $f^*$ preserves post-fixed points.*

*Proof.* Assume $h_B$ is a post-fixed point of $T_B$, i.e., $h_B \preceq T_B(h_B)$. By order preservation, $f^* h_B \preceq f^* T_B(h_B)$. By laxity, $f^* T_B(h_B) \preceq T_A(f^* h_B)$. By transitivity, $f^* h_B \preceq T_A(f^* h_B)$, which means that $f^* h_B$ is a post-fixed point of $T_A$.  □

**Lemma C.15.** *If the closure operators are oplax along an order-reflecting pullback $f^*$, then $f^*$ reflects post-fixed points.*

*Proof.* Assume $f^* h_B$ is a post-fixed point of $T_A$, i.e., $f^* h_B \preceq T_A(f^* h_B)$. By oplaxity, $T_A(f^* h_B) \preceq f^* T_B(h_B)$. By transitivity, $f^* h_B \preceq f^* T_B(h_B)$. By order reflection, $h_B \preceq T_B(h_B)$, which means that $h_B$ is a post-fixed point of $T_B$.  □

Summarizing the above results, we obtain the following theorem.

**Theorem C.16.** *Given an $F$-coalgebra homomorphism $f : (A, t_A) \to (B, t_B)$, a lifting $\lambda$, and the closure operators $T_A$ and $T_B$, we have $T_A \circ f^* \preceq f^* \circ T_B$, and*

- *if the lifting $\lambda$ is strict at $f$, then $T_A \circ f^* = f^* \circ T_B$, and the pullback $f^*$ preserves post-fixed points;*
- *if $f^n$ is an epimorphism, then the pullback $f^*$ reflects post-fixed points.*

*Proof.* The oplaxity of the closure along pullback is a direct result of Lemma C.9 and Lemma C.10. The post-fixed point preservation is due to Lemmas C.7, C.10 and C.14. The post-fixed point reflection is due to Lemmas C.8 and C.15.  □

# D. Pushforward

We now introduce the notion of pushforward of bundles, which is the left adjoint of pullback.

**Definition D.1** (Pushforward bundle). A *pushforward* of $n$-ary $V$-bundles along a **C**-morphism $f : A \to B$ is a left adjoint $f_* : H_V^n A \to H_V^n B$ of the pullback $f^* : H_V^n B \to H_V^n A$ in Def. C.2,

$$H_V^n A \underset{f^*}{\overset{f_*}{\underset{\perp}{\rightleftarrows}}} H_V^n B, \tag{91}$$

which satisfies the following adjunction property:

$$\forall h_A : A^n \to V.\ \forall h_B : B^n \to V.\ (f_* h_A \preceq h_B) \leftrightarrow (h_A \preceq f^* h_B). \tag{92}$$

$$\begin{array}{ccc} A^n & \xrightarrow{\ f^n\ } & B^n \\ & \underset{h_A}{\searrow} \ \overset{\preceq}{\phantom{x}} \ \underset{f_* h_A}{\swarrow} & \\ & V & \end{array} \tag{93}$$

In our pointwise setting, the pushforward bundle can be computed as joins over the fibers of $f$:

$$\begin{aligned} f_* :\quad H_V^n A &\to & H_V^n B \\ h_A &\mapsto & \left[ b_{1:n} \mapsto \bigvee_{a_i \in f^{-1}(b_i)} h_A(a_{1:n}) \right] \end{aligned} \tag{94}$$

A basic property of pushforward and pullback is given by the following unit and counit equations of the adjunction.

**Lemma D.2** (Unit and counit). *Under Def. D.1, for every $f : A \to B$ we have:*

$$\mathrm{id}_{H_V^n A} \preceq f^* \circ f_*. \tag{95a}$$

$$\forall h_A : A^n \to V.\ h_A \preceq f^* f_* h_A. \tag{95b}$$

$$f_* \circ f^* \preceq \mathrm{id}_{H_V^n B}. \tag{96a}$$

$$\forall h_B : B^n \to V.\ f_* f^* h_B \preceq h_B. \tag{96b}$$

*Proof.* Eq. (95) is the *unit* of the adjunction, obtained by instantiating Eq. (92) with $h_B = f_* h_A$; Eq. (96) is the *counit* of the adjunction obtained by instantiating Eq. (92) with $h_A = f^* h_B$. $\square$

## D.1. Pushforward of order

Similarly to Lemma C.7, we can show that the pushforward is order-preserving.

**Lemma D.3.** *The pushforward $f_*$ is order-preserving.*

*Proof.* Given two bundles $h_A, h'_A : A^n \to V$, assume $h_A \preceq h'_A$. By the unit, $h_A \preceq h'_A \preceq f^* f_* h'_A$. By the adjunction, this is equivalent to $f_* h_A \preceq f_* h'_A$, which means that $f_*$ is order-preserving. $\square$

## D.2. Pushforward of pullback

**Lemma D.4.** *Given an $F$-coalgebra homomorphism $f : (A, t_A) \to (B, t_B)$, we have*

$$\begin{array}{ccc} H_V^n F A & \xrightarrow{\ (Ff)_*\ } & H_V^n F B \\ {\scriptstyle t_A^*}\downarrow & \preceq & \downarrow{\scriptstyle t_B^*} \\ H_V^n A & \xrightarrow{\ f_*\ } & H_V^n B \end{array} \tag{97a}$$

$$f_* \circ t_A^* \preceq t_B^* \circ (Ff)_*. \tag{97b}$$

$$\forall h_A : A^n \to V.\ f_* t_A^* h_A \preceq t_B^* (Ff)_* h_A. \tag{97c}$$

*Proof.* By Lemma D.2, we have $t_A^* \preceq t_A^* \circ (Ff)^* \circ (Ff)_*$. By Lemma C.9, we have $t_A^* \circ (Ff)^* = f^* \circ t_B^*$. Composing them yields $t_A^* \preceq f^* \circ t_B^* \circ (Ff)_*$, which is equivalent to $f_* \circ t_A^* \preceq t_B^* \circ (Ff)_*$ by adjunction. $\square$

### D.3. Pushforward of lifting

**Lemma D.5.** *Given an $F$-coalgebra homomorphism $f : (A, t_A) \to (B, t_B)$ and a lifting $\lambda$, we have*

$$
\begin{array}{ccc}
H_V^n A & \xrightarrow{\ f_* \ } & H_V^n B \\
{\scriptstyle \lambda_A} \downarrow & \preceq & \downarrow {\scriptstyle \lambda_B} \\
H_V^n F A & \xrightarrow{\ (Ff)_* \ } & H_V^n F B
\end{array}
\tag{98a}
$$

$$
(Ff)_* \circ \lambda_A \preceq \lambda_B \circ f_*. \tag{98b}
$$

$$
\forall h_A : A^n \to V. \ (Ff)_* \lambda_A(h_A) \preceq \lambda_B(f_* h_A). \tag{98c}
$$

*Proof.* By Lemma D.2, we have $\lambda_A \preceq \lambda_A \circ f^* \circ f_*$. By Lemma C.10, we have $\lambda_A \circ f^* \preceq (Ff)^* \circ \lambda_B$. Composing them yields $\lambda_A \preceq (Ff)^* \circ \lambda_B \circ f_*$, which is equivalent to $(Ff)_* \circ \lambda_A \preceq \lambda_B \circ f_*$ by adjunction. $\square$

### D.4. Pushforward of closure

As in Def. C.13, we introduce the notion of laxity and oplaxity along pushforward.

**Definition D.6** (Lax and oplax along pushforward). Given $F$-coalgebras $(A, t_A)$ and $(B, t_B)$, a morphism $f : A \to B$, a lifting $\lambda$ of $n$-ary bundles, and the closure operators $T_A$ and $T_B$ induced by the lifting and the coalgebras, we say that the closure operators are *lax* (resp. *oplax*) *along pushforward* $f_*$ if the following diagram commutes up to $\preceq$ (resp. $\succeq$):

$$
\begin{array}{ccc}
H_V^n A & \xrightarrow{\ f_* \ } & H_V^n B \\
{\scriptstyle \lambda_A} \downarrow & & \downarrow {\scriptstyle \lambda_B} \\
H_V^n F A & \preceq & H_V^n F B \\
{\scriptstyle t_A{}^*} \downarrow & & \downarrow {\scriptstyle t_B{}^*} \\
H_V^n A & \xrightarrow{\ f_* \ } & H_V^n B
\end{array}
\tag{99a}
$$

$$
f_* \circ T_A \preceq T_B \circ f_*. \tag{99b}
$$

$$
\forall h_A : A^n \to V. \ f_* T_A(h_A) \preceq T_B(f_* h_A). \tag{99c}
$$

We can then establish the following result about the preservation of post-fixed points along pushforward.

**Lemma D.7.** *If the closure operators are lax along an order-preserving pushforward $f_*$, then $f_*$ preserves post-fixed points.*

*Proof.* Assume $h_A$ is a post-fixed point of $T_A$, i.e., $h_A \preceq T_A(h_A)$. By order preservation, $f_* h_A \preceq f_* T_A(h_A)$. By laxity, $f_* T_A(h_A) \preceq T_B(f_* h_A)$. By transitivity, $f_* h_A \preceq T_B(f_* h_A)$, which means that $f_* h_A$ is a post-fixed point of $T_B$. $\square$

The dual statement also holds and is omitted here.

Finally, we can establish the following theorem summarizing the results on pushforward of closure.

**Theorem D.8.** *Given an $F$-coalgebra homomorphism $f : (A, t_A) \to (B, t_B)$, a lifting $\lambda$, and the closure operators $T_A$ and $T_B$, we have $f_* \circ T_A \preceq T_B \circ f_*$, and the pushforward $f_*$ preserves post-fixed points.*

*Proof.* The laxity of the closure along pushforward is a direct result of Lemma D.4 and Lemma D.5. The post-fixed point preservation is due to Lemmas D.3 and D.7. $\square$

# E. Proofs

This appendix restates the main transfer theorems and records short proof sketches that expose the argument structure without repeating the detailed category-theoretic proofs. The supporting ingredients are the lifting results in Appendix B, the pullback closure theorem in Appendix C, and the pushforward closure theorem in Appendix D. We then include the detailed proofs of the Section 4 propositions.

**Theorem 3.12** (Safe verification). *If $\phi : (S, t_S) \to (Z, t_Z)$ is a surjective homomorphism, then the pullback $\phi^*$ reflects post-fixed points: for any bundle $h_Z : Z^n \to V$,*

$$(h_Z \preceq T_Z(h_Z)) \leftarrow (\phi^* h_Z \preceq T_S(\phi^* h_Z)). \tag{21}$$

*Proof sketch.* We first establish the oplaxity of closure along pullback. The pullback-of-pullback identity in Lemma C.9 and the pullback-of-lifting inequality in Lemma C.10 give the abstract closure comparison; for the coupling-based probability component used in our binary liftings, the needed oplaxity premise is supplied by Proposition B.23. Applied to the encoder $\phi$, this yields

$$T_S(\phi^* h_Z) \preceq \phi^* T_Z(h_Z). \tag{100}$$

If $\phi^* h_Z \preceq T_S(\phi^* h_Z)$, then by transitivity $\phi^* h_Z \preceq \phi^* T_Z(h_Z)$. Next, surjectivity makes $\phi^*$ order-reflecting by Lemma C.8, and Lemma C.15 turns these two facts into reflection of post-fixed points. This is summarized in Theorem C.16, so we conclude $h_Z \preceq T_Z(h_Z)$. $\qquad\square$

**Theorem 3.13** (Safe construction). *If $\phi : (S, t_S) \to (Z, t_Z)$ is a homomorphism, then the pushforward $\phi_*$ preserves post-fixed points: for any bundle $h_S : S^n \to V$,*

$$(h_S \preceq T_S(h_S)) \to (\phi_* h_S \preceq T_Z(\phi_* h_S)). \tag{22}$$

*Proof sketch.* We first establish the laxity of closure along pushforward. The adjunction in Lemma D.2, the pullback–pushforward interaction in Lemma D.4, and the pushforward-of-lifting inequality in Lemma D.5 together yield

$$\phi_* T_S(h_S) \preceq T_Z(\phi_* h_S). \tag{101}$$

If $h_S \preceq T_S(h_S)$, then order preservation of pushforward in Lemma D.3 gives

$$\phi_* h_S \preceq \phi_* T_S(h_S). \tag{102}$$

Combining the two displays and using transitivity yields $\phi_* h_S \preceq T_Z(\phi_* h_S)$, and Lemma D.7 identifies this as preservation of post-fixed points. This is summarized in Theorem D.8. $\qquad\square$

**Theorem 3.14** (Relation of logical and quantitative bundles). *The zero predicate $z : [0, \infty] \to \{\top, \bot\} := [x \mapsto (x = 0)]$ maps a quantitative bundle $h_X : X^n \to [0, \infty]$ to a logical bundle $z \circ h_X : X^n \to \{\top, \bot\}$. If a quantitative lifting $\lambda_X^{[0,\infty]}$ and a logical lifting $\lambda_X^{\{\top,\bot\}}$ are constructed with operators in Table 2 such that the zero predicate $z$ is an algebra homomorphism between the operators,[9] then*

$$z \circ \lambda_X^{[0,\infty]}(h_X) = \lambda_X^{\{\top,\bot\}}(z \circ h_X). \tag{23}$$

*Proof sketch.* This is an instance of commutative lifting in Def. B.26. The assumptions say that the zero predicate $z$ is an algebra homomorphism for the three operator building blocks in Table 2, so the combinator, aggregator, and barycenter all commute with $z$. By the morphism-level characterization in Eq. (76b), the induced quantitative and logical liftings therefore commute with postcomposition by $z$. Applying that identity to a bundle $h_X$ gives

$$z \circ \lambda_X^{[0,\infty]}(h_X) = \lambda_X^{\{\top,\bot\}}(z \circ h_X). \tag{103}$$

Thus thresholding the quantitative lifting at zero agrees pointwise with lifting the corresponding logical bundle. $\qquad\square$

**Proposition 4.1.** *If $f : X \to Y$ is a homomorphism from $\langle t_X, o_X \rangle$ to $\langle t_Y, o_Y \rangle$, then it is also a homomorphism from $(\mathbb{P}o_X)^A \circ t_X : X \to (\mathbb{P}O)^A$ to $(\mathbb{P}o_Y)^A \circ t_Y : Y \to (\mathbb{P}O)^A$, i.e., $\forall x \in X. \forall a \in A. o_{Y*} t_Y(f(x), a) =_{\mathbb{P}O} o_{X*} t_X(x, a).$*

*Proof.* The diagram in Eq. (7a) is equivalent to

$$
\begin{array}{ccc}
X \xrightarrow{\quad f \quad} Y & & X \xrightarrow{\quad f \quad} Y \\
{\scriptstyle t_X}\downarrow \qquad \downarrow {\scriptstyle t_Y} & \text{and} & {\scriptstyle o_X}\downarrow \qquad \downarrow {\scriptstyle o_Y} \\
(\mathbb{P}X)^A \xrightarrow{(\mathbb{P}f)^A} (\mathbb{P}Y)^A & & O \xrightarrow{\quad \mathrm{id}_O \quad} O
\end{array}
\tag{104}
$$

---

[9]This means that (i) $z \circ \oplus_{[0,\infty]} = \oplus_{\{\top,\bot\}} \circ (z \times z)$, (ii) $z \circ \square_{[0,\infty]} = \square_{\{\top,\bot\}} \circ z^A$, and (iii) $z \circ \odot_{[0,\infty]} = \odot_{\{\top,\bot\}} \circ \mathbb{P}z$.

The desired result is immediate from these two diagrams:

$$
\begin{array}{ccc}
X & \xrightarrow{\;\;\;\;\;f\;\;\;\;\;} & Y \\
{\scriptstyle t_X}\downarrow & & \downarrow{\scriptstyle t_Y} \\
(\mathbb{P}X)^A & \xrightarrow{\;(\mathbb{P}f)^A\;} & (\mathbb{P}Y)^A \\
{\scriptstyle (\mathbb{P}o_X)^A}\downarrow & & \downarrow{\scriptstyle (\mathbb{P}o_Y)^A} \\
(\mathbb{P}O)^A & \xrightarrow{\;\mathrm{id}_{(\mathbb{P}O)^A}\;} & (\mathbb{P}O)^A
\end{array}
\tag{105}
$$

□

**Proposition 4.2.** *A model-irrelevance abstraction in* Li et al. *(2006) is exactly a coalgebra homomorphism in Def.* 2.6.

*Proof.* The model-irrelevance abstraction condition can be rewritten as: $\ker\phi \to \ker(\phi_* t_S) \wedge \ker o_S$, where the consequent is exactly $\ker(F_{\mathrm{Moore}}\phi \circ \langle t_S, o_S\rangle)$. Thus, $\ker\phi \to \ker(F_{\mathrm{Moore}}\phi \circ \langle t_S, o_S\rangle)$ means that $F_{\mathrm{Moore}}\phi \circ \langle t_S, o_S\rangle : S \to F_{\mathrm{Moore}}Z$ factors through $\phi : S \to Z$; equivalently, there exists a morphism $\langle t_Z, o_Z\rangle : Z \to F_{\mathrm{Moore}}Z$ such that $F_{\mathrm{Moore}}\phi \circ \langle t_S, o_S\rangle = \langle t_Z, o_Z\rangle \circ \phi$. This is exactly the coalgebra homomorphism condition for $F_{\mathrm{Moore}}$ in Eq. (7). □

**Proposition 4.3.** *For an equivalence* $r : X \times X \to \{\top, \bot\}$, *let* $q_r : X \to X/r$ *be its quotient map. For an* $F$-*coalgebra* $(X, t_X)$, *define a monotone operator* $T_X$ *on equivalences as follows:*

$$
T_X(r) := \ker(F q_r \circ t_X) = t_X{}^* \ker(F q_r).
\tag{24}
$$

*Then, the quotient map* $q_r$ *is a homomorphism if and only if the equivalence relation* $r$ *is a post-fixed point of* $T_X$.

*Proof.* If $r$ is a post-fixed point of $T_X$, then $r = \ker q_r \to \ker(F q_r \circ t_X)$. This means that $F q_r \circ t_X : X \to F(X/r)$ factors through $q_r : X \to X/r$, i.e., there exists a morphism $t_{X/r} : X/r \to F(X/r)$ such that $F q_r \circ t_X = t_{X/r} \circ q_r$. This is exactly the coalgebra homomorphism condition in Def. 2.6. Conversely, if there exists such a coalgebra structure $t_{X/r}$, then $F q_r \circ t_X = t_{X/r} \circ q_r$ implies that $\ker q_r \to \ker(F q_r \circ t_X)$, i.e., $r$ is a post-fixed point of $T_X$. □

# F. Discussion

This section clarifies how several nearby RL constructions sit relative to our framework. Coalgebra homomorphisms compare systems of the same type and induce state abstraction. By contrast, interface changes, closed-loop feedback, temporal abstraction, and agent memory often change the system type or wire systems together before any abstraction map is chosen.

## F.1. Functorial transformations of systems

We first make explicit the categorical mechanism behind Section 2.3. A natural transformation changes the system type uniformly on each carrier, and therefore induces a functor between the corresponding coalgebra categories (Rutten, 2000, Theorem 15.1):

**Lemma F.1.** *Given two endofunctors $F, G : \mathbf{C} \to \mathbf{C}$ on a category $\mathbf{C}$, a natural transformation $\alpha : F \Rightarrow G$ induces a functor $\mathbf{C}_\alpha : \mathbf{C}_F \to \mathbf{C}_G$ between the corresponding coalgebra categories:*

$$\begin{array}{cccc}
\mathbf{C}_\alpha : & \mathbf{C}_F & \to & \mathbf{C}_G \\
& (A, a : A \to FA) & \mapsto & (A, \alpha_A \circ a : A \xrightarrow{a} FA \xrightarrow{\alpha_A} GA) \\
& f : (A, a) \to (B, b) & \mapsto & f : (A, \alpha_A \circ a) \to (B, \alpha_B \circ b)
\end{array} \tag{106}$$

$$\begin{array}{ccc}
A & \xrightarrow{\ f\ } & B \\
{\scriptstyle a}\downarrow & & \downarrow{\scriptstyle b} \\
FA & \xrightarrow{\ Ff\ } & FB \\
{\scriptstyle \alpha_A}\downarrow & & \downarrow{\scriptstyle \alpha_B} \\
GA & \xrightarrow{\ Gf\ } & GB
\end{array} \tag{107}$$

However, not all functors between coalgebra categories are induced by natural transformations. Being induced by a natural transformation is a strong condition: it forces the carrier to stay the same and the change of structure to be uniform.

The examples below use this distinction to separate changes of system type from state abstraction. They transform the type of system while keeping the carrier fixed, before any state abstraction is added. We proceed from uniform interface changes to transformations that involve closed-loop feedback or agent memory.

## F.2. Basic interface transformations

A simple example is the *action translation* functor, which changes the input interface of the system by translating actions from one set to another:

**Example F.2** (Action translation). An action translation $g : A \to A'$ transforms a system $\mathsf{t} : S \to S^{A'}$ into a system $S^g \circ \mathsf{t} : S \to S^A$:

$$\begin{array}{c}
S \to \boxed{\mathsf{t}} \to S \\
\uparrow \\
\cdots\cdots A' \cdots\cdots \\
\uparrow \\
\boxed{g} \\
\uparrow \\
A
\end{array} \tag{108}$$

A functor between coalgebra categories is given by

$$\begin{array}{cccc}
& \mathbf{C}_{(-)^{A'}} & \to & \mathbf{C}_{(-)^A} \\
& (S, \mathsf{t} : S \to S^{A'}) & \mapsto & (S, S^g \circ \mathsf{t} : S \to S^A) \\
& f : (S, \mathsf{t}) \to (S', \mathsf{t}') & \mapsto & f : (S, S^g \circ \mathsf{t}) \to (S', S^g \circ \mathsf{t}')
\end{array} \tag{109}$$

where $S^g : S^{A'} \to S^A := (-) \circ g$ is the precomposition map.

The new system has the same carrier $S$, but presents a different action interface. When the translated system receives an action $a \in A$, it runs the original system with action $g(a) \in A'$. Thus the map $g$ can identify several actions of the new interface, or leave some actions of the original interface unused. In either case, no states are merged; only the way the system consumes actions has changed. Thus action translation changes the input interface uniformly. It is the simplest interface adapter that will reappear in the discussion of MDP homomorphisms.

### F.3. Closing the loop with a policy

Another basic transformation closes an input interface by feeding back an action chosen from the current observation.

**Example F.3** (Closed-loop system). A policy $\pi : O \to A$ transforms an open-loop system $\langle t, o \rangle : S \to S^A \times O$ into a closed-loop system $p_\pi(t, o) : S \to S \times O$:

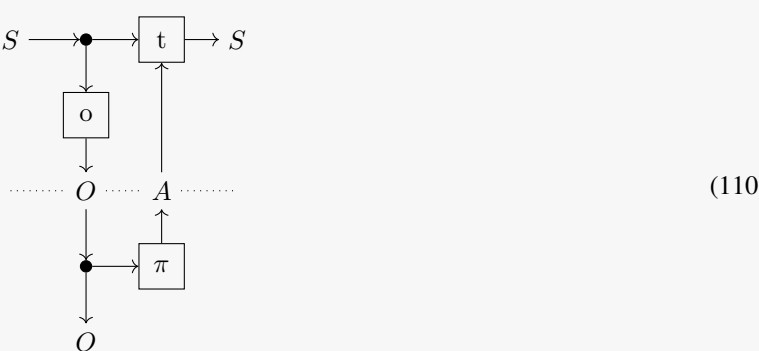

$$(110)$$

A functor between coalgebra categories is given by

$$
\begin{array}{ccc}
\mathbf{C}_{(-)^A \times O} & \to & \mathbf{C}_{- \times O} \\
(S, \langle t, o \rangle : S \to S^A \times O) & \mapsto & (S, p_\pi(t, o) : S \to S \times O) \\
f : (S, \langle t, o \rangle) \to (S', \langle t', o' \rangle) & \mapsto & f : (S, p_\pi(t, o)) \to (S', p_\pi(t', o'))
\end{array}
\tag{111}
$$

where

$$
p_\pi(t, o) := S \xrightarrow{\langle \mathrm{id}_S, o \rangle} S \times O \xrightarrow{\mathrm{id}_S \times \langle \pi, \mathrm{id}_O \rangle} S \times (A \times O) \xrightarrow{\cong} (S \times A) \times O \xrightarrow{t \times \mathrm{id}_O} S \times O,
\tag{112}
$$

$$
= S \xrightarrow{\langle t, o \rangle} S^A \times O \xrightarrow{\mathrm{id}_{S^A} \times \langle \pi, \mathrm{id}_O \rangle} S^A \times (A \times O) \xrightarrow{\cong} (S^A \times A) \times O \xrightarrow{\epsilon_S \times \mathrm{id}_O} S \times O,
\tag{113}
$$

and $\epsilon_S : S^A \times A \to S$ is the evaluation map.

This is the construction used in Section 2 to obtain closed-loop systems from open-loop systems and policies. Together, action translation and closed-loop feedback give the two ingredients needed for the observation-dependent action translation below.

### F.4. State-action abstraction: MDP homomorphism

One relevant notion in the RL literature is *MDP homomorphism* (Ravindran & Barto, 2001; 2002), which consists of a state map $S \to S'$ together with a state-dependent action map $S \times A \to A'$. This concept has been used for *state-action abstraction* (Ravindran & Barto, 2003; 2004; Ravindran, 2004; Taylor et al., 2008; van der Pol et al., 2020a;b; Rezaei-Shoshtari et al., 2022; Panangaden et al., 2024; Bakirtzis et al., 2025). This is not the same construction as the coalgebra homomorphism in Def. 2.6. A coalgebra homomorphism compares systems of the same type, so the input-output interface, including the action and observation spaces, remains fixed. An MDP homomorphism combines state abstraction with an interface adapter. We separate these two components in this paper: the abstraction map is handled by coalgebra homomorphisms, while the action map induces a natural transformation between systems of different types.

The following example records the interface adapter. It combines the two basic transformations above: as in action translation, it changes the action interface; as in closed-loop feedback, the translation can depend on the current output observation. When the observation is the full state, this specializes to the state-dependent action maps used in MDP homomorphisms.

**Example F.4** (Observation-dependent action translation). An observation-dependent action translation $g : O \times A \to A'$ transforms a system $\langle \mathrm{t}, \mathrm{o} \rangle : S \to S^{A'} \times O$ into a system $p_g(\mathrm{t}, \mathrm{o}) : S \to S^A \times O$:

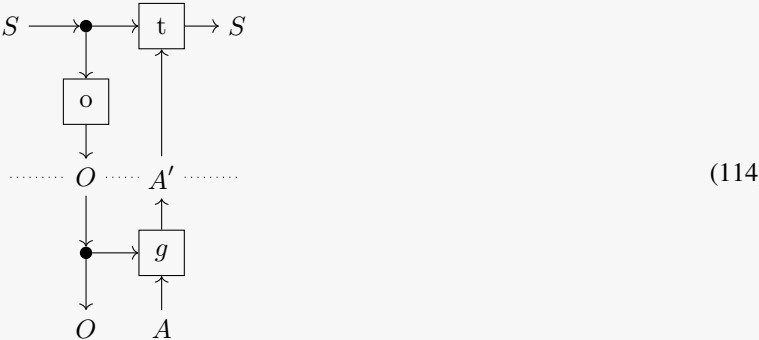

$$(114)$$

A functor between coalgebra categories is given by

$$
\begin{array}{ccc}
\mathbf{C}_{(-)^{A'} \times O} & \to & \mathbf{C}_{(-)^{A} \times O} \\
(S, \langle \mathrm{t}, \mathrm{o} \rangle : S \to S^{A'} \times O) & \mapsto & (S, p_g(\mathrm{t}, \mathrm{o}) : S \to S^A \times O) \\
f : (S, \langle \mathrm{t}, \mathrm{o} \rangle) \to (S', \langle \mathrm{t}', \mathrm{o}' \rangle) & \mapsto & f : (S, p_g(\mathrm{t}, \mathrm{o})) \to (S', p_g(\mathrm{t}', \mathrm{o}'))
\end{array}
\tag{115}
$$

where

$$
p_g(\mathrm{t}, \mathrm{o}) := S \xrightarrow{\langle \mathrm{t}, \mathrm{o} \rangle} S^{A'} \times O \xrightarrow{\mathrm{id}_{S^{A'}} \times \langle g, \mathrm{id}_O \rangle} S^{A'} \times (A'^A \times O) \xrightarrow{\cong} (S^{A'} \times A'^A) \times O \xrightarrow{\circ \times \mathrm{id}_O} S^A \times O,
\tag{116}
$$

and $\circ : S^{A'} \times A'^A \to S^A$ is the composition map.

Thus the observation-dependent action translation functor can be read as the interface transformation part of MDP homomorphisms (Ravindran & Barto, 2001; 2002). Combined with a coalgebra homomorphism on the state carrier, it gives the two ingredients of state-action abstraction without conflating them.

Relatedly, since pullback is simply precomposition, given a policy $\pi_Z : Z \to A$ on the representation space $Z$, we can pull it back along the encoder $\phi : S \to Z$ to obtain a policy $\phi^* \pi_Z : S \to A := \pi_Z \circ \phi$ on the concrete state space $S$. This *pullback policy* construction has been used in Ravindran & Barto (2001; 2002); Rezaei-Shoshtari et al. (2022). Together with a *section* $S \times A' \to A$ (a right inverse of the action map $S \times A \to A'$ in the second component), it was referred to by Panangaden et al. (2024, Definition 7) as a *lifted policy* in the context of MDP homomorphism. Note that this "*lifting*" is an RL term for concretizing abstract actions and is unrelated to the bundle lifting discussed in Section 3.2.

### F.5. State augmentation: forward statistic aggregation

The previous examples modify how an environment exposes or consumes its interface. We next turn to state augmentation: an observation-producing system is paired with a forward statistic state that is updated chronologically from the observations it emits.

**Example F.5** (Forward statistic aggregation). Let $M$ be a set of auxiliary statistic states. A statistic update map $\rhd : O \times M \to M$ updates such a statistic state from a new observation and transforms a system $\langle \mathrm{t}, \mathrm{o} \rangle : S \to S \times O$ into an augmented system $p(\mathrm{t}, \mathrm{o}, \rhd) : S \times M \to S \times M$:

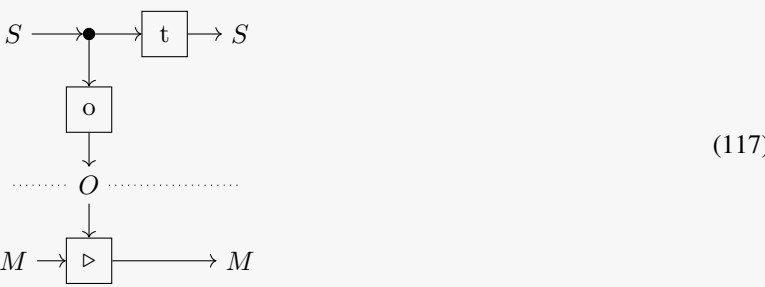

$$(117)$$

A bifunctor between coalgebra categories is given by

$$
\begin{array}{ccccc}
\mathbf{C}_{-\times O} & \times & \mathbf{C}_{(-)^O} & \to & \mathbf{C}_{\mathrm{id}_{\mathbf{C}}} \\
(S, \langle \mathrm{t}, \mathrm{o} \rangle : S \to S \times O) & & (M, \rhd : M \to M^O) & \mapsto & (S \times M, p(\mathrm{t}, \mathrm{o}, \rhd) : S \times M \to S \times M) \\
f : (S, \langle \mathrm{t}, \mathrm{o} \rangle) \to (S', \langle \mathrm{t}', \mathrm{o}' \rangle) & & f_\rhd : (M, \rhd) \to (M', \rhd') & \mapsto & f \times f_\rhd : (S \times M, p(\mathrm{t}, \mathrm{o}, \rhd)) \to (S' \times M', p(\mathrm{t}', \mathrm{o}', \rhd'))
\end{array}
\tag{118}
$$

where

$$
p(\mathrm{t}, \mathrm{o}, \rhd) := S \times M \xrightarrow{\langle \mathrm{t}, \mathrm{o} \rangle \times \mathrm{id}_M} (S \times O) \times M \xrightarrow{\cong} S \times (O \times M) \xrightarrow{\mathrm{id}_S \times \rhd} S \times M,
\tag{119}
$$

$$
= S \times M \xrightarrow{\langle \mathrm{t}, \mathrm{o} \rangle \times \rhd} (S \times O) \times M^O \xrightarrow{\cong} S \times (M^O \times O) \xrightarrow{\mathrm{id}_S \times \epsilon_O} S \times M,
\tag{120}
$$

and $\epsilon_O : M^O \times O \to M$ is the evaluation map.

This forward statistic aggregation functor combines two systems: one that produces observations and one that updates a statistic from them. The observation may contain more information than this statistic update needs; any extraction of the update-relevant part is left implicit in $\rhd$. Unlike the backward reward-aggregation closure in Example 3.4, this is a forward, chronological construction: it augments the carrier from $S$ to $S \times M$ and updates the second component along the generated observation stream. Examples include belief-state filters in POMDPs (Kaelbling et al., 1998), predictive state representations (Littman & Sutton, 2001; Singh et al., 2004), temporal monitors and reward machines (Camacho et al., 2019; Icarte et al., 2018; 2022), and budget or safety-accounting statistics (Altman, 1999; Wachi et al., 2024). In all of these cases, the statistic records information about the observation history, but it does not feed actions back into the environment in this construction.

### F.6. Stateful agent

Forward statistic aggregation already carries an auxiliary statistic state, but that state is passive with respect to the environment input: it receives observations and does not choose actions. A stateful agent uses memory in both directions: observations update the memory, and the memory produces future actions, as shown below (cf. Fig. 1):

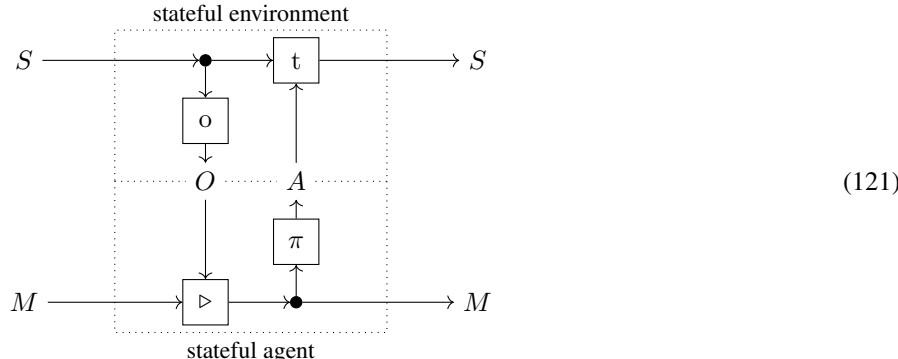

$$\tag{121}$$

The environment carries state $S$, evolves according to the selected action, and emits observations in $O$. The agent carries memory $M$, updates that memory from each observation, and uses the updated memory to choose future actions. This makes the agent state an explicit part of the closed-loop system rather than an implicit implementation detail of the policy.

In classical finite-state controller architectures, this move corresponds to the need for non-reactive policies, reducing the problem to learning finite-state automata (Meuleau et al., 1999); see also Amato et al. (2010) for variations using Mealy rather than Moore automata, and Koul et al. (2019) for implementations using RNNs. Belief states in POMDPs (Kaelbling et al., 1998) provide a different, yet consistent view, where these memory states correspond to beliefs about environment states and are updated by Bayesian filtering. Predictive state representations (Littman & Sutton, 2001) go further by making no explicit assumptions about the environment state space. They generate memory states and transitions by compressing finite histories of action-observation pairs, a process that can be seen as approximating causal states and their transitions, giving $\epsilon$-transducers of bi-infinite stochastic channels (Rosas et al., 2025). This aligns with the critique of the "*environment spotlight*" in Abel et al. (2024): if the formal model only exposes the environment state, then agent state and adaptation are easy to hide inside the policy.

Note that this differs from forward statistic aggregation, where the observation-producing system feeds a separate statistic

update, giving a one-way construction that augments the carrier with auxiliary history. In a stateful agent, the memory is not only a record of past observations: it also determines future actions, and those actions affect the future environment state and future observations. Thus the relevant closed system contains both environment state and agent memory, and abstractions may need to preserve behavior that depends on their interaction. This issue appears, for instance, in bisimulations for MDPs, which often require rewards (Givan et al., 2003; Ravindran & Barto, 2003; 2004; Ravindran, 2004; Taylor et al., 2008) or value (Givan et al., 2003; Li et al., 2006) to be preserved so that the same actions or policies are produced. It also appears for POMDPs, which are either transformed into a belief MDP first and then coarse grained using standard MDP bisimulations (Castro et al., 2009), or coarse grained directly with extensions of MDP bisimulations that preserve rewards (Rosas et al., 2025). These treatments, however, remain mostly feed-forward, in the sense that they simply package properties of the environment into an abstraction for an agent to use under certain constraints. In this sense, the reciprocal feedback described above is not captured by the one-way functorial construction used for forward statistic aggregation. A fully categorical treatment of this wiring would likely require a more general formalism, such as categorical systems theory with wiring diagrams, lenses, or optics where the outputs of each component can be connected to the inputs of the other, and a double categorical treatment that generalizes natural transformations to squares, or 2-cells; see for instance (Myers, 2023; Capucci, 2025; Libkind & Myers, 2025). We leave that extension to future work.

### F.7. Temporal abstraction: hierarchical reinforcement learning and options

The classical *options* framework formalizes *temporal abstraction* by adding temporally extended courses of action to a primitive MDP (Sutton et al., 1999; Precup, 2000). An option is defined by a triple: the states where the option can be initiated, the intra-option policy over primitive actions, and the termination condition (Sutton et al., 1999, Section 2). An option can be selected only at states in its initiation set. Once selected, it persists across primitive time steps. The intra-option policy chooses primitive actions, the environment evolves, and after each new state or observation the termination condition, the agent decides whether control remains with the same option or returns to a policy over options.

In our framework, the active option can be treated as a part of an agent state. Thus options fit the stateful-agent picture above by specializing the agent memory $M$ to a set of option indices $I$, as shown below:

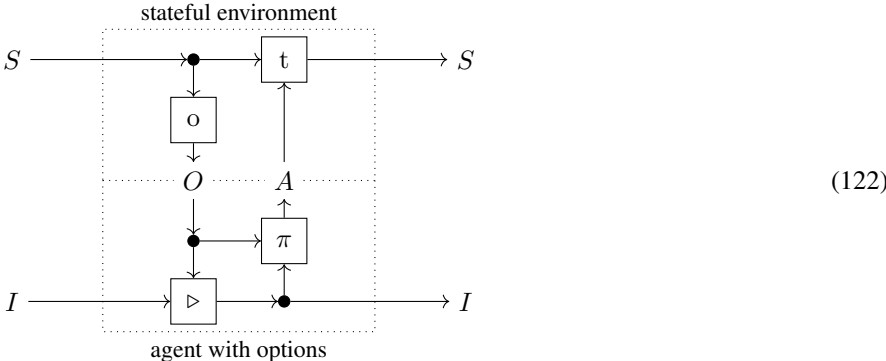

$$(122)$$

Following the index-based formulation (Klissarov et al., 2025), the policy becomes an intra-option policy $\pi : S \times I \to \mathbb{P}A$, while the update map $\triangleright : O \times I \to I$ summarizes how options terminate and are selected:

$$o \triangleright i := \begin{cases} \text{a new option index} & \text{if option } i \text{ terminates given observation } o, \\ \text{option } i & \text{otherwise.} \end{cases} \qquad (123)$$

This option-selection rule may itself be learned, so it can be a part of the agent rather than part of the environment dynamics. When observations are full states, this recovers the usual Markov option formulation. With richer histories, it matches the semi-Markov generalization developed by Precup (2000).

The essential point is that the option index is not an environment state, but internal control state that indexes a subpolicy. Fixing $i$ gives the primitive policy $\pi(-, i)$ until the update switches to another index. This is the same mechanism used by the thought states of Hanna & Corrado (2025): the internal state selects which state-dependent policy is currently executed, while changes to that internal state do not directly change the environment state. The macro-action or SMDP view is therefore a quotient description of this closed-loop execution. One runs the primitive system until the active option terminates, then records the resulting state, elapsed duration, and accumulated reward. SubMDP and hierarchy-decomposition methods

are closely related, which pursue the same temporal abstraction direction by exposing regions, exits, or other environment structure (Hengst, 2002; Wen et al., 2020). We leave further investigation of the connections between state abstraction and temporal abstraction to future work.

