# OpenReview forum: "Compositional Behavioral Semantics for State Abstraction in Reinforcement Learning"
_ICML.cc/2026/Conference — ICML 2026 regular_

### Official Review · Reviewer_2EzM · 2026-03-13

**Soundness:** 4
**Presentation:** 4
**Significance:** 3
**Originality:** 4
**Overall Recommendation:** 4
**Confidence:** 1

**Summary:**

This paper proposes a unifying framework for studying behavior preservation under state abstraction in reinforcement learning using tools from coalgebra. State abstraction is crucial in RL because real-world problems often involve very large state spaces, making it necessary to reduce the dimensionality of the state representation in order to make learning and planning tractable. Existing works study state abstraction from different perspectives, typically focusing on preserving specific behavioral aspects of the system, such as value functions, optimal policies, or bisimulation relations. This work proposes a foundational theoretical framework for state abstraction based on coalgebraic representations of dynamical systems. In this framework, RL environments are modeled as coalgebras, and the behavioral properties that one wishes to preserve under abstraction (e.g., value functions or bisimulation relations) are formalized as behavioral structures, represented as bundles. State abstraction is defined as a homomorphism between systems, which ensures that system dynamics are preserved under the abstraction. Using this formulation, the authors provide a unified perspective for analyzing how different behavioral structures transfer between concrete and abstract systems.

**Compliance With Llm Reviewing Policy:**

Affirmed.

**Key Questions For Authors:**

- The paper claims that the framework may help in designing novel training objectives. In what sense is the framework generative? Does it provide systematic procedures for constructing new behavioral structures or training objectives?
- Might this framework enable the derivation of new behavioral metrics or abstraction criteria from the unified formulation, beyond reproducing existing concepts such as bisimulation metrics?
- The framework appears to connect behavioral structures with logical specifications. For example, the paper mentions logical predicates and safety properties as instances of behavioral structures. This suggests potential connections with areas such as symbolic reinforcement learning or temporal logic reward specifications. Could the authors elaborate how this framework could be used to study such symbolic or neurosymbolic reward specifications, or how symbolic specifications could benefit from it?

**Limitations:**

yes

**Strengths And Weaknesses:**

Weaknesses
- While the proposed framing appears elegant and mathematically well structured, the implications or the derivation of new objectives proposed in future extension is not immediately clear. At present, the work seems primarily foundational, providing a unified perspective on existing abstraction theories. It would be useful to better understand whether the framework can lead to new behavioral metrics or abstraction criteria, or whether its primary role is to organize and unify existing approaches to state abstraction.

Strengths
- I find the paper well written and carefully structured. The proposed unified framework is mathematically elegant and introduces a principled and generic approach to reasoning about behavioral preservation in state abstraction.
- The paper presents a general theoretical framework that unifies several previously separate concepts in the state abstraction literature under a single formal perspective. This coalgebraic formulation provides a clean and principled way to reason about different behavioral preservation criteria and may serve as a useful theoretical lens for studying abstraction in reinforcement learning.

---

> ### Author Rebuttal · Authors · 2026-03-29
>
> We thank the reviewer for the positive assessment and the thoughtful questions about the downstream payoff of the framework.
>
> We addressed the concerns as follows.
>
> ```
> Generative value: "In what sense is the framework generative?"
> ```
>
> The generative aspect is at the level of specification: by choosing a bundle arity, a codomain, and lifting operators, one determines the behavioral structure under study.
>
> For example, in `footnote 5` after `Examples 3.3 and 3.4`, we briefly mentioned that there are other barycenter operators in addition to the almost-sure operator (e.g., positive probability: *it is at least possible*; threshold: *with probability greater than $\theta$*).
> We can also use operators such as median or quantiles instead of the expectation operator to define other more conservative/permissive or robust quantitative behavioral evaluation.
> The framework does not just collect known notions; it parameterizes them in a common space of choices.
> Some concrete examples can be found in `Wachi et al., (2024)`.
>
> Under the proposed framework, we can properly ask questions such as *"if two states have similar values, does it imply (or how much does it imply) that they have similar safety risks, and does the guarantee hold after abstraction?"*
> This is the sense in which the framework is intended to be constructive, not only unifying.
>
> As the focus of this paper is on developing the general toolkit rather than specific case studies, we leave further investigations for future work.
> We will extend the discussion and include more concrete examples to further illustrate how this toolkit can be used.
>
> Another potential use of the proposed framework is to analyze "**stateful agents**".
> In this paper, to align with the existing studies of state abstraction, we focused on the environment modeled as a stochastic Moore machine, with actions $A$ as input and observations $O$ as output; while the agent is simply a "stateless policy" $O \to A$.
> As the environments and tasks become more complex, it is natural to equip agents with some internal states to keep track of the learning/planning/task-solving processes.
> Therefore, a stateful agent can also be modeled as a stochastic Moore machine, with observations $O$ as input and actions $A$ as output (other options available, see `footnote 1`).
>
> We can study "**agent state abstraction**" using the proposed behavioral structure definition and proof techniques, and model a **closed-loop system** with both environment state and agent state evolving using a natural transformation.
> Note that `Theorems 3.11--3.13` are parameterized by the functor (system type) and lifting (behavioral structure), so they still hold for more complex systems.
>
> We believe that adopting this algebraic perspective can generate new research opportunities and deeper insights.
> We will include such discussions in the revised version.
>
> > Abel et al., "Three Dogmas of Reinforcement Learning." Reinforcement Learning Conference. 2024. https://openreview.net/forum?id=XlI0ccgpP3
>
> ```
> New metrics or criteria: "Might this framework enable the derivation of new behavioral metrics or abstraction criteria?"
> ```
>
> Yes, at the level of construction principles rather than as a fixed catalogue of objectives.
> We will strengthen the discussion around `Theorem 3.13` to emphasize that it gives a recipe: once a logical behavioral structure is specified as a bundle and the operators satisfy the homomorphism condition, the corresponding quantitative structure follows automatically.
> We agree that this point deserves fuller examples, and we plan to expand the worked examples around `Theorem 3.13`.
>
> ```
> Symbolic specifications: "Could the authors elaborate ... symbolic or neurosymbolic reward specifications?"
> ```
>
> This is a natural use case of the logical side of the framework.
>
> Let us take a closer look at `Example 3.3`: the safety predicate on observations $h_O: O \to \\{\top, \bot\\}$ is not necessarily a given function; it can be a neural network trained on images or videos.
> Given $h_O$, we can define a behavioral structure with our desired meaning.
> According to `Theorem 3.13`, we can design a quantitative (and sometimes even differentiable) metric, which can serve as a learning objective in a neurosymbolic system, or evaluate the quality of the safety predicate $h_O$.
> `Theorems 3.12 and 3.13` (together with `Proposition 4.2`) can predict how this behavioral structure should behave under abstraction.
>
> In this work, we only introduced a small fraction of coalgebraic tools.
> A fuller treatment of temporal logic or neurosymbolic objectives is beyond the scope of the current paper, but this is exactly the kind of extension the logical-bundle formulation is meant to support.
> We believe some soundness, completeness, and decidability results and other coalgebraic modal logic techniques `(Rutten, 2000; Kurz, 2001; Pattinson, 2003)` can be useful, and we are excited to bring them to the RL community to study concrete RL problems.

---

> > ### Author Rebuttal · Reviewer_2EzM · 2026-04-03
> >
> > I thank the authors for their response. I am curios about the future directions of this foundational work, so I will maintain my positive score.

---

> > > ### Author Response · Authors · 2026-04-06
> > >
> > > Thank you for your positive feedback and encouraging assessment of this foundational work!
> > > We look forward to using the proposed framework to study directions such as
> > > - agent state abstractions,
> > > - approximate guarantees via lax homomorphisms,
> > > - behavioral-structure-preserving abstractions beyond strict homomorphisms, and
> > > - symbolic or temporal-logic specifications.
> > >
> > > We will include a brief discussion of these directions in the future work section.

---

### Official Review · Reviewer_NWDJ · 2026-03-16

**Soundness:** 3
**Presentation:** 2
**Significance:** 4
**Originality:** 3
**Overall Recommendation:** 5
**Confidence:** 3

**Summary:**

The authors present a category theoretic framework for abstraction and simulation for reinforcement learning. The framework uses an F-coalgebra to define a state transition functorially. With this definition, the authors cast abstraction and transitions under a policy as homomorphisms and natural transformations, respectively. They use the concepts of bundling and lifting to formalize the notions of behavioral structure, which are used in the main theoretical results, demonstrating the requirements for verification on abstract systems, constructing behavioral structures on abstract systems, and relating quantitative semantics to qualitative semantics. The authors conclude with examples from recent literature, cast in the category theoretic language they have proposed.

**Compliance With Llm Reviewing Policy:**

Affirmed.

**Final Justification:**

This is overall a good paper, and the authors adequately addressed my concerns.

**Key Questions For Authors:**

See above

**Limitations:**

Yes

**Strengths And Weaknesses:**

This paper is timely and relevant to many members of the ICML community. The goals are worthwhile—developing a unified view of abstraction and simulation for RL is likely to prove quite valuable.

This paper is quite dense, and it makes for a difficult read. This paper has the unenviable task of introducing very abstract mathematics to an audience that may not have much facility with category theory. Writing such a paper in a manner that is self-contained is surely difficult, and the authors have done a good job overall in distilling the essentials of category theory, including ample appendices. Nonetheless, it may be a struggle for readers who come to this paper without a very narrow set of expertise.

The primary weaknesses of this paper are in presentation and clarity. Because of the page limit, the authors have put much of the technical definitions in the appendix. The result is a dense, notation-heavy paper. If the authors could add more connective text to help orient the reader with some intuition, it would go a long way.

The examples provided with Moore machines are helpful, assuming that the audience is familiar with them. Still, the examples could be improved a bit, even if they are helpful. For example, providing a simple definition, or even a picture of a Moore machine would be helpful for grounding the example for a reader who is otherwise unfamiliar.

The examples stop after Section 3.2. It would be very helpful to continue examples. Otherwise, the value of the theorems as well as the semantics in Sec 4 may be hard to discern for the unfamiliar reader.

Some notation and terminology is used before it is introduced. For example $\mathbb{P}S$ is used on line 78, but not defined until 120. Similarly, pushforward is used around line 120 but not formally introduced until much later. The authors should take care to make sure all notation and definitions are properly introduced.

---

> ### Author Rebuttal · Authors · 2026-03-29
>
> We thank the reviewer for the encouraging assessment and the concrete suggestions on accessibility.
> We agree that it is challenging to present such a math-dense work to the general RL audience, and your suggestions help us further improve the readability.
>
> We plan to address the issues as follows.
>
>
> ```
> Readability: "This paper is quite dense." "The primary weaknesses of this paper are in presentation and clarity."
> ```
>
> According to your suggestion, we will add more **connective text**, **intuitive explanations**, **pointers**, and **short roadmap sentences** at the start of the main technical sections or after introducing new concepts, so readers can track the role of each section/definition before the notation accumulates.
> We will also add a brief bridge in the `Section 3.5` to remind the reader which concrete structures are covered by the general statements.
>
> The appendix/main-text split was a page-limit tradeoff, so in revision, we will focus on adding orientation in the main body rather than expanding the formal development.
>
> ```
> Examples and scaffolding: "The examples stop after Section 3.2."
> ```
>
> Near the first Moore machine example (`Examples 2.3 and 2.4`), we will add a short intuitive description for readers who do not already know the terminology.
>
> We will point out the stochastic Moore machine and hidden Markov model examples in `Figure 1`, which diagrammatically shows their input-output interface.
>
> We will also carry the running examples into `Sections 3.3--3.5 and 4`, so the transfer theorems and the `Section 4` instantiations are introduced as continuations of the same safety predicate, value function, and bisimulation relation/metric thread rather than as isolated case studies.
>
> ```
> Notation: "Some notation and terminology are used before being introduced."
> ```
>
> Thank you for pointing it out.
> We will move the explanation for $\mathbb{P}$ to where it first appears.
>
> We agree that the overloaded **pushforward** notation needed gentler signposting.
>
> - In `l.120`, $f_*$ or $\mathbb{P}f: \mathbb{P}X \to \mathbb{P}Y$ is the usual **pushforward of distributions**/**pushforward measure**, which is a standard concept in probability theory.
> - We only defined **pushforward of bundles** in `Definition 3.9` (denoted as $f_*: C^{X^n} \to C^{Y^n}$ or $f_*: \mathrm{Hom}(X^n, C) \to \mathrm{Hom}(Y^n, C)$, to be more precise), which is a related but different construction.
>
> In $f\_\*p$, if $p \in \mathbb{P}X$ is a distribution, we can infer $f\_\*$ is the pushforward of distributions; while in $f\_\* h\_X$, if $h\_X: X^n \to C$ is a bundle, we can infer $f\_\*$ is the pushforward of bundles.
>
> The pullback $f^\*$ (contravariant, from target to source) and pushforward $f\_\*$ (covariant, from source to target) terminology and notation are consistent with their uses in other fields of mathematics.
> We will clarify their differences and relations in the revised version.

---

> > ### Author Rebuttal · Reviewer_NWDJ · 2026-04-02
> >
> > Acknowledged.

---

> > > ### Author Response · Authors · 2026-04-06
> > >
> > > Thank you for acknowledging our rebuttal!

---

### Official Review · Reviewer_br6J · 2026-03-20

**Soundness:** 3
**Presentation:** 2
**Significance:** 3
**Originality:** 4
**Overall Recommendation:** 4
**Confidence:** 4

**Summary:**

This paper aims to unify research on state abstractions using the language of category theory. Specifically, the authors formalise the environment as a stochastic Moore machine functor, the agent as a hidden Markov model functor, and the agent–environment interaction loop as a natural transformation between the two. This leads to the notion of a behavioural structure, defined as the post-fixed point of a closure operator (for example, the value-function fixed point of a Bellman operator). Within this theoretical framework, the paper then analyses various properties of state abstractions, including which behavioural structures are preserved under abstraction. The framework is also used to re-derive standard results in the literature. In particular, the authors show that homomorphic encoders (state-abstraction maps) preserve the next-observation distribution, and that abstraction notions such as model-irrelevance and bisimulation can be expressed in terms of homomorphisms.

**Compliance With Llm Reviewing Policy:**

Affirmed.

**Final Justification:**

Thank you to the authors for the detailed feedback. I'm happy with the provided clarifications and promised paper revisions. I have increased my score to a weak accept in recognition of the significance this work (unifying state abstraction theory) despite its limitations (non-exhaustive replication of current guarantees, and new guarantees are left to future works).

**Key Questions For Authors:**

Please refer to the weaknesses above. For example:

* Where can the proofs of the main theorems be found? I am assuming I missed them in the original submission, otherwise I unfortunately will have to drop my score to a reject.
* Can the authors clarify what important nuances am I missing that makes the given theorems particularly insightful? If stronger/novel consequences of the framework can be demonstrated, this would increase my assessment of the paper’s significance.
* Does the framework allow one to establish necessary (rather than only sufficient) conditions for the preservation or transfer of behavioural structures, such as optimal policies or value functions and convergence of a standard learning algorithms (e.g. policy gradients, QL)? Demonstrating such results would substantially strengthen the contribution.
* What assumptions are required on the state, action, and observation spaces (e.g. finiteness, compactness, measurability)? Clarifying this would improve both soundness and reproducibility.
* Can the authors more clearly distinguish between general category-theoretic constructions and their specific instantiations in RL? It currently seems like an structured mixture of both. E.g. Definitions 2.2. and 2.6. are defining general category theory terms like system and homomorphism using RL specific terms like dynamics (the stated ambiguity "we consider coalgebra, system, and dynamic as synonyms" severely hurst clarity).  A clearer separation would significantly improve readability and perceived novelty.

**Limitations:**

Yes

**Strengths And Weaknesses:**

## Strengths

* The idea of using category theory to unify different notions of state abstraction and analyse their properties is interesting and timely. I also appreciate the use of multiple illustrative examples that help clarify aspects of the presentation.
* The overall theoretical formulation appears broadly sound, although there are some mathematical imprecision and presentation issues. I also really like the framing of the agent-environment interaction as a natural transformation, and the definition of behavioural structures as the post-fixed point of a closure operator.
* The paper provides several theoretical statements (mainly three theorems and propositions), together with an extensive appendix with additional theoretical results. Notably,  it shows that standard known results regarding model-irrelevance and bisimulation abstractions can be rederived using the proposed framework.

## Weaknesses

### Major
* None of the main results (theorems) have easily identifiable proofs (I couldn't find them). They do not appear in the “Proofs” section (Appendix F), and searching for the theorem numbers or names yields no results. The main text also does not provide proof sketches.
* Additionally, the claims of these theorems unfortunately appear somewhat underwhelming given the substantial effort devoted to the coalgebra framing. For example, Theorem 3.11 states that a surjective encoder preserving transition dynamics also preserves the ordering over behavioural structures (e.g. value functions). This result seems relatively straightforward even without category-theoretic machinery, unless there is an important nuance that is not clearly communicated. Similar concerns apply to the other theorems.
* The only other main theoretical results in the main paper whose proofs are clearly provided are Propositions 4.1-4.3. While these are relevant and usefull, they largely restate known results and therefore offer limited new insight.
* The paper would have benefited from deriving genuinely novel insights using the introduced machinery. For instance, it would be valuable to analyse whether a surjective homomorphic encoder is a *necessary* condition for transferring behavioural structures (e.g. preserving optimal policies), or more generally to provide necessary and/or sufficient conditions for the transfer of concrete behavioural structures. Likewise, the framework could be used to provide new guarantees for concrete abstraction methods in the literature that currently lack theoretical justification.
* Equation (7c.i') does not appear well defined, since the map $f$ may not be invertible. A similar issue arises in Definition 3.9, unless $f$ is intended to be bijective (which should be stated explicitly).
* The paper states “we also equip (C) with an order $\preceq$”, but does not clarify whether this is meant to be a partial order, a total order, or a well-order. It seems that the generality of the results require a partial order, but this should be specified.

### Minor
* The presentation is difficult to follow and could be significantly improved. A substantial portion of the paper is devoted to redefining general category-theoretic structures (often informally in the main text, with formal definitions relegated to the appendix), rather than instantiating them directly in the RL setting that is the focus of the work. This makes it unclear which definitions are standard textbook material and which are novel contributions of the paper. It also obscures when the discussion is meant to be general versus specific to the agent-environment interaction.
* The mathematical formulations are often imprecise.
  * The paper does not clearly state assumptions on the state space (S), action space (A), or observation space (O). For example, it is unclear whether these are finite sets, or whether compactness or measurability assumptions (e.g. Borel structure) are required.
  * The notation is sometimes inconsistent. The symbols (X) and (Y) are introduced as arbitrary sets in general definitions, but are also used in ways that appear to represent state spaces. It is unclear whether (S) and (X) denote the same set, and what assumptions distinguish (X) and (Y) from (S).
  * $v^\pi$ in Equation (16) is undefined. I assume it is the value function, but be explicit. Given the writing, I think it should actually be an arbitrary mapping from states to reals, such that the fixed point of the Bellman operator is the value function.
  * In Definition 3.7, $x_1$, ..., $x_n$ should be in brackets.
  * The pushforward operator is defined in Section 3 but used before it (e.g. in example 2.3.).
  * Equation (7c.i) should use commas instead of dots to separate the universal statements
  * In page 3: "A core requirement in state abstraction is to preserve the dynamics of the original system in the abstract system.". Is it true though? How about value-preserving or optimal-policy preserving ones?

---

> ### Author Rebuttal · Authors · 2026-03-29
>
> We thank the reviewer for the careful reading.
>
> ## Major issues
>
> ```
> Proof location: "I couldn't find the proofs of the main theorems."
> ```
>
> In the current version, we did provide categorical statements and proofs for `Lemma 3.10` and `Theorems 3.11--3.13` in `Appendices C--E` and a proof roadmap in the first paragraph of `Appendix F`. That said, we agree that in the original submission, the proofs were scattered and not as visible as they should have been.
>
> To improve accessibility, in revision, `Appendix F` now restates the main theorems and adds a short proof sketch below each theorem, with explicit pointers to the supporting results, enabling readers to grasp the essence of the proof technique without reading through the technical details in previous appendices.
>
> For example, we sketch `Theorem 3.11` as follows:
>
> > We first establish the oplaxity of closure along pullback. Theorem D.16 combines the pullback-of-pullback identity with the pullback-of-lifting inequality; for the coupling-based probability component in the binary lifting, the needed premise is Proposition C.23. Applied to the encoder $\phi$, this gives $T_S(\phi^* h_Z) \preceq \phi^* T_Z(h_Z)$. If $\phi^* h_Z \preceq T_S(\phi^* h_Z)$, then $\phi^* h_Z \preceq \phi^* T_Z(h_Z)$. Since $\phi$ is surjective, $\phi^*$ reflects order, so $h_Z \preceq T_Z(h_Z)$.
>
> ```
> Technical precision: "Equation (7c.i') does not appear well defined" etc.
> ```
>
> We addressed these issues directly:
>
> - **Inverse image**: Under `Eq. (7c.i')` (`l.165`) and in `Definition 3.9` (`l.293`), we denoted $f^{-1}(y)$ as the `inverse image` (a set). We further emphasize this by saying it is a `fiber (inverse image)`, not a functional inverse.
> - **Preorder**: `Section 3.1` now states that bundles are compared using a `preorder` $\preceq$.
> - **Assumptions**: We do not have any finiteness or measurability assumptions on sets $S$, $A$, or $O$, leading to a minor limitation that only  *finitely supported distributions* are allowed (`l.120`, so that the exponential object/function set ${(\mathbb{P}X)}^A$ is properly defined). We clarified this detail and leave extensions to future work.
>
> ```
> Why the framework matters: "The claims of the theorems ... appear underwhelming. What makes them particularly insightful?"
> ```
>
> The intended contribution is not that each transfer theorem is surprising in isolation, but that the paper gives a parameterized definition-and-proof schema for behavioral structures.
> The parameters are the bundle arity, codomain, and lifting operators, and the same transfer theorems then apply across predicates, values, relations, and metrics.
>
> Concretely, `Proposition 4.2` shows that model-irrelevance / self-prediction abstraction is exactly the homomorphism condition (not articulated in the literature), and `Proposition 4.3` shows that these homomorphisms are quotient maps of bisimulation equivalences.
> Thus, once a method learns a homomorphic abstraction, `Theorems 3.11 and 3.12` provide guarantees for a broad class of behavioral structures, not only the one used to motivate the objective.
> This also helps explain why self-prediction objectives are attractive in practice, aligning with recent empirical investigations of these objectives (see also `Ni et al., 2024`; `Luo et al., 2025`).
>
> Please see also our reply to `Reviewer 2EzM`.
>
> ```
> Scope: "Does the framework allow one to establish necessary conditions?"
> ```
>
> As the focus of this paper is on developing the general toolkit, it does not establish necessary conditions.
> However, the general framework can be used to systematically establish an *order/hierarchy* of specific behavioral structures (a la `Li et al., 2006`) to provide new guarantees for more flexible abstractions, based on the lifting operators, structure decomposition, or algorithm-specific designs.
> We noted this future work direction (*behavioral structure-preserving maps*), and we will expand the discussion to clarify the current scope and the intended next step.
>
> ## Other issues
>
> - $X$, $Y$ vs. $S$, $Z$: $X$ and $Y$ are arbitrary sets while $S$ (state space) and $Z$ (representation space) are sets with special meanings in abstraction. We will include a table of symbols in the appendix.
> - The **Bellman** example now writes the operator on an arbitrary $v$, with $v^\pi$ named only as its fixed point.
> - $\mathbb{P}f$ or $f_*$ is the usual **pushforward of distributions**. The later **pushforward of bundles** is a distinct covariant construction on bundles. The shared name and notation follow the standard pullback/pushforward convention for transport along a map, as in topology, geometry, and probability.
> - In this paper, dynamics is intended as informal language for the one-step behavior encoded by a coalgebra (`Rutten, 2000`), not as an RL-specific term. But we agree that the wording around **dynamics** could blur the distinction between the formal and informal levels, and we will tighten it in the final revision.
> - Other small issues will also be fixed.

---

> > ### Author Rebuttal · Reviewer_br6J · 2026-04-06
> >
> > Thank you to the authors for the detailed feedback. I'm happy with the provided clarifications and promised paper revisions, and will increase my score accordingly.

---

> > > ### Author Response · Authors · 2026-04-06
> > >
> > > Thank you again for your careful follow-up.
> > > We are pleased that our clarifications addressed your concerns and that you are now considering raising your score.
> > > We will ensure that all of your suggestions are incorporated into the revised version.

---

### Decision · Program_Chairs · 2026-04-30

**Decision:**

Accept (regular)

**Comment:**

The reviewers agreed that this paper provides an original and timely unified framework for understanding state abstraction in reinforcement learning through the lens of co-algebraic category theory. While reviewers initially raised valid concerns regarding the density of the presentation, the visibility of key mathematical proofs, and the immediate generative value of the framework, the authors provided extensive rebuttals that incorporated reviewer feedback to improve readability and highlight the framework's practical parameterization. Given its technical soundness, foundational significance, and the clear consensus among reviewers that this work will serve as a valuable, reusable toolkit for the broader ICML community, I recommend acceptance.